# Integrated land and water-borne geophysical surveys shed light on the sudden drying of large karst lakes in southern Mexico

Matthias Bücker[1,2], Adrián Flores Orozco[2], Jakob Gallistl[2], Matthias Steiner[2], Lukas Aigner[2], Johannes Hoppenbrock[1], Ruth Glebe[1], Wendy Morales Barrera[3], Carlos Pita de la Paz[4], César Emilio García García[4], José Alberto Razo Pérez[4], Johannes Buckel[1], Andreas Hördt[1], Antje Schwalb[5], Liseth Pérez[3,5]

1Institute of Geophysics and Extraterrestrial Physics, TU Braunschweig, Braunschweig, 38106, Germany.
2Department of Geodesy and Geoinformation, Research Unit Geophysics, TU Wien, Vienna, 1040, Austria.
3Instituto de Geología, Universidad Nacional Autónoma de México, Mexico City, 04510, Mexico.
4Geotem Ingeniería S.A. de C.V., Mexico City, 14640, Mexico.
5Institute of Geosystems and Bioindication, TU Braunschweig, Braunschweig, 38106, Germany.

*Correspondence to*: Matthias Bücker, (m.buecker@tu-braunschweig.de)

**Abstract.** Karst water resources play an important role in drinking water supply, but are highly vulnerable to even slight changes in climate. Thus, solid and spatially dense geological information is needed to model the response of karst hydrological systems to such changes. Additionally, environmental information archived in lake sediments can be used to understand past climate effects on karst water systems. In the present study, we carry out a multi-methodological geophysical survey to investigate the geological situation and sedimentary infill of two karst lakes (Metzabok and Tzibaná) of the Lacandon Forest in Chiapas, southern Mexico. Both lakes present large seasonal lake-level fluctuations and experienced an unusually sudden and strong lake-level decline in the first half of 2019, leaving Lake Metzabok (maximum depth ~25 m) completely dry and Lake Tzibaná (depth ~70 m) with a water level decreased by approx. 15 m. Before this event, during a lake-level high stand in March 2018, we collected water-borne seismic data with a sub-bottom profiler (SBP) and transient electromagnetic (TEM) data with a newly-developed floating single-loop configuration. In October 2019, after the sudden drainage event, we took advantage of this unique situation and carried out complementary measurements directly on the exposed lake floor of Lakes Metzabok and Tzibaná. During this second campaign, we collected time-domain induced polarization (TDIP) and seismic refraction tomography (SRT) data. By integrating the multi-methodological data set, we (1) identify 5-6 m thick, likely undisturbed sediment sequences on the bottom of both lakes, which are suitable for future paleoenvironmental drilling campaigns, (2) develop a comprehensive geological model implying a strong interconnectivity between surface water and karst aquifer, and (3) evaluate the potential of the applied geophysical approach for the reconnaissance of the geological situation of karst lakes. This methodological evaluation reveals that under the given circumstances, (i) SBP and TDIP phase images consistently resolve the thickness of the fine-grained lacustrine sediments covering the lake floor, (ii) TEM and TDIP resistivity images consistently detect the upper limit of the limestone bedrock and the geometry of fluvial deposits of a river delta, and (iii) TDIP and SRT images suggest the existence of a layer that separates the lacustrine sediments from the limestone bedrock and consists of collapse debris mixed with lacustrine sediments. Our results show that the combination of seismic methods, which are most widely used for lake-bottom reconnaissance, with resistivity-based methods such as TEM and TDIP can

significantly improve the interpretation by resolving geological units or bedrock heterogeneities, which are not visible from
seismic data. Only the use of complementary methods provides sufficient information to develop comprehensive geological
models of such complex karst environments

## 1 Introduction

About 7–12% of the world's continental area is covered by karst (e.g., Hartmann et al., 2014) and up to one quarter of the
earth's population at least partially depends on drinking water from karst systems (e.g., Ford and Williams, 2007). Even though
continued population growth and industrialization put pressure on these important resources in terms of both water quantity
and quality, the response of karst systems to expected future climate change is still not well understood (Hartmann et al., 2014).
Groundwater models offer one opportunity to estimate future changes in water availability, but heavily depend on reliable and
spatially dense geological information. Where direct geological information, e.g., from drillings, is not dense enough or not
available at all, geophysical methods can be used to provide quasi-continuous indirect information on the subsurface geology
in karst areas (Bechtel et al., 2007).

Another possibility to understand climatic effects on karst water systems relies on the analysis of paleoenvironmental records
(e.g., Medina-Elizalde and Rohling, 2012; Vázquez-Molina et al., 2016). In particular, lake sediments are important archives
of freshwater and terrestrial environmental information and sediment cores can be used to reconstruct past climate and
ecological changes in the lakes (Cohen, 2003; Schindler, 2009; Pérez et al., 2020). Thus, paleoenvironmental studies give
insight into the local links between climate variations and the availability (and quality) in lakes and the connected karst aquifer
system. To identify suitable drilling locations providing continuous paleoenvironmental records at a high temporal resolution,
knowledge about sediment thickness and composition, depth to bedrock, and possible heterogeneities within the lake sediments
is needed (Last and Smol, 2002).

Geophysical methods can efficiently provide such information from the local scale up to the lake-basin scale and can
(principally) be employed on both land and water. Due to the usually sharp contrast between seismic velocities of sediment
layers and the underlying bedrock, (reflection-) seismic methods are often given priority over other geophysical methods for
lake-bottom reconnaissance (Scholz, 2002). In particular, low-frequency echo sounders (e.g., Dondurur, 2018), also referred
to as sub-bottom profilers (SBP), allow to quickly map sediment deposits of several tens of meters based on single-channel
seismic data. Nevertheless, electrical-resistivity images provided by electrical resistivity tomography (e.g., Binley and Kemna,
2005) or electromagnetic soundings (e.g., Kaufman et al., 2014) complement the mostly geometrical information obtained
from reflection-seismic or sub-bottom profiling measurements (Butler, 2009). Under certain conditions such as high lake-bed
reflectivity and/or low reflectivity of targeted boundaries, seismic methods may, however, provide insufficient results and
therefore alternative methods are needed.

Recent studies using direct-current (DC) electrical resistivity for water-borne investigations on freshwater bodies include
surveys with floating (e.g., Befus et al., 2012; Orlando, 2013; Colombero et al., 2014) or underwater electrode chains (e.g.,

Toran et al., 2015) and provide evidence for the potential of this method for shallow-water applications. Electrical resistivity can also be assessed by electromagnetic methods, which, compared to DC resistivity measurements, offer a more compact experimental layout. Electromagnetic surveys are often carried out as transient electromagnetic (TEM) soundings with floating magnetic sources and receivers. Hatch et al. (2010), for example, used an in-loop configuration with a ~7.5 m x 7.5 m transmitter and a ~2.5 m x 2.5 m receiver to map river bed salinization in an Australian river with an average water depth of 5-10 m. Mollidor et al. (2013) used a similar but slightly larger setup (~18 m x 18 m transmitter, ~6 m x 6 m receiver) to map a thick conductive sediment layer below the bottom of a 20-m deep maar lake in Germany. More recently, Yogeshwar et al. (2020) used the system developed by Mollidor et al. to image a volcanic lake hydrothermal system on the Azores; whereas Lane et al. (2020) introduced a compact floating TEM system, designed for the rapid electrical mapping of the subsurface of rivers and estuaries. Some older relevant case studies with shallow-water applications of both techniques, DC resistivity and electromagnetic soundings, were reviewed by Butler (2009).

In a previous study, we successfully used geoelectrical and electromagnetic methods to investigate the sedimentary infill of two desiccated lakes in a volcanic area (Bücker et al., 2017; Lozano-García et al., 2017). To extend our investigations, in this study, we evaluate the potential of land and water-borne resistivity-imaging methods to complement seismic methods for the investigation of karst lakes in the Lacandon Forest, southern Mexico. Recent biological and abiotic studies have highlighted the great potential of sedimentary sequences from the lakes of this remote area as continuous paleoenvironmental and paleoclimatic records during the late Quaternary (e.g., Díaz et al., 2017; Echeverría-Galindo et al., 2019; Charqueño-Celis et al., 2020). In this study, we focus on Lakes Metzabok and Tzibaná, two of the largest lakes of the Lacandon Forest, which experienced a sudden and catastrophic lake-level drop in the first half of 2019. While large seasonal lake-level variations are part of the nature of both lakes, it remains unclear whether such particular events as the one observed in 2019, which left Lake Metzabok completely dry, are also recurrent with a frequency of several decades or rather linked to recent climate change. To better understand possible draining mechanisms and their triggers, besides further paleoenvironmental investigations, a comprehensive geological picture of the lakes´ geological situation is essential.

In 2018, roughly one year before the drainage event and when the lakes were filled, we collected seismic data with a SBP and carried out TEM soundings to assess the electrical resistivity of the lake bottom and obtain information on the thickness of the sedimentary infill. The sudden drainage of the investigated lakes in 2019, provided us with the unique opportunity to collect additional data directly on the dry lake bed. Seismic data was then recollected with a seismic refraction tomographic (SRT) setup in order to provide information on both refractor geometry and seismic velocities of the different geophysical units. Additional electrical imaging data was measured with the time-domain induced polarization (TDIP) method, which has fewer limitations regarding the detectability of thin near-surface layers and heterogeneities than the transient electromagnetic method. Furthermore, the polarization properties of the subsurface materials assessed by TDIP measurements provide additional information and can improve the interpretation of TEM and TDIP resistivity results.

Based on the above considerations, our study has three main objectives: (1) Identify suitable drilling locations to obtain undisturbed and far-reaching sedimentary sequences for paleoenvironmental reconstructs, (2) provide basic knowledge on the

geological situation of the studied lakes (sediment cover, limestone bedrock and possible connectivity with the karst aquifer), and (3) develop and apply a multi-methodological geophysical approach with a special focus on the evaluation of the potential of water-borne TEM soundings for lake-bottom reconnaissance.

## 2 Study area

The study area is located in the Lacandon Forest (16°–17.5° N; 90.5°–92° W; 500–1500 m.a.s.l.), which occupies the north-eastern part of the State of Chiapas, Mexico (Fig. 1a). The region belongs to the Chiapas fold belt with its WNW-trending folds and thrusts, which mainly developed in massive cretaceous limestone (García-Gil and Lugo Hupb, 1992). The orogeny of the Chiapas fold belt is related to the collision of the Tehuantepec Transform/Ridge on the Cocos plate with the Middle America Trench during the Middle Miocene (Mandujano-Velazquez and Keppie, 2009). The resulting anticlines and synclines dominate the topography in the study area forming long WNW-directed valleys and mountain ranges. The tectonically fractured limestone geology, in conjunction with the humid subtropical climate, favour an intensive karstification (García-Gil and Lugo Hupb, 1992). In the valleys, lakes formed by bedrock dissolution, such as dolines (or sinkholes), uvalas (formed by two or more dolines) and poljes (larger karst depressions), are mostly aligned in the main fold direction.

The lake system of Metzabok (17°6'30"–17° 8'30" N, 91°36'30"– 91°38'50" W, ~550 m a.s.l.) consists of 21 lakes of different sizes, the majority of which are interconnected when water levels are high (Lozada Toledo, 2013). The two largest lakes of the system are Lake Tzibaná (area 1.24 km²; max. depth 70 m) and Lake Metzabok (0.83 km²; 25 m) (see Fig. 1b). The river Nahá is the principal superficial tributary connecting the lake system of Metzabok with the one of Nahá (~830 m a.s.l.); a superficial outlet of the lake system does not exist. Although the (additional) water supply and discharge through the underlying karst system is unknown, fast lake-level changes indicate substantial groundwater-surface water connections. Usually, seasonal lake-level changes amount to ~10 m and can be traced back to prehispanic times (Lozada Toledo, 2013). Between March and August 2019 an extreme lake-level drop occurred that left Lake Metzabok completely dry and decreased the water level of Lake Tzibaná by ~15 m.

## 3 Data acquisition and processing

With the primary goal of mapping sediment thicknesses below the lake floor of various lakes of the karst lake systems of Metzabok and Nahá, we carried out a first geophysical campaign employing seismic (SBP) and TEM methods, when lake levels were maximum in March 2018 (Fig. 2a). Immediately after the dramatic lake-level decline, we revisited the study site in October 2019 to collect refraction seismic tomography (SRT) data and perform TDIP measurements directly on the dry lake bottom (Fig. 2a, b).

## 3.1 Electrical resistivity measurements in the laboratory

In October 2019, a total of six surface sediment samples (top 10 cm) and two water samples were collected at different locations
for laboratory analyses (see sampling locations in Fig. 2a, b). On the dry lake bottom, sediment samples were collected using
a small spade, whereas an Ekman grab sampler was used to retrieve sediment samples from water-covered areas. Sediment
samples were stored in sealed plastic bags in order to prevent the loss of moisture; water samples were stored in plastic bottles.
All samples were kept cool during transport and storage in order to prevent an increased degradation of organic matter. The
electrical conductivity of the water samples (at 20°C) was measured with a laboratory probe. The frequency-dependent
complex electrical resistivity of the samples was measured using a Chameleon I measuring device (Przyklenk et al., 2016) in
the frequency range from 1 mHz to 240 kHz. To this end, the unconsolidated sediments were filled into four-point measuring
cells with non-polarizing potential electrodes as used by Kruschwitz (2007) and Bairlein et al. (2014). Prior to and during the
measurement, the measuring cell was stored in a climate chamber to keep the sample at a constant temperature of 20°C.
Measurements were repeated over a period of 4 to 5 days after filling the cell and inserting it into the climate chamber in order
to assure equilibrium conditions in the sample. Measurements on relatively dry samples (MET19-A and TSI19-A) resulted in
comparably high phase values. These samples were removed from the measuring cell, saturated with water of the
corresponding lake (using one of the two water samples), and filled again into the measuring cell. This procedure led to more
consistent phase measurements compared to the other samples.

## 3.2 Collection of sub-bottom profiler (SBP) lines

Low-frequency echo-sounders, often referred to as sub-bottom profilers (SBP), are single channel seismic reflection systems,
which are used to obtain bathymetric profiles and provide a high-resolution stratigraphic display of the uppermost sediments
(e.g., Dondurur, 2018). In March 2018, SBP lines were collected with the 10-kHz transducer StrataBox HD (SyQwest), which
has an output power of 300 W, mounted on a motor boat. Data was recorded with a record length of 200 ms and a 1024 Hz
sampling frequency. The SBP device was mounted mid-ships in a side mount configuration, with the transducer positioned at
0.4 m below the water surface. Prior to each survey, the acoustic wave velocity profiles of the water columns of the two studied
lakes were measured with a Digibar S (Odom Hydrographic). SBP lines were laid out in a regular NS- and EW-oriented grid
with separations of 100 m and 300 m, respectively (see Fig. 1b). Navigation data was measured with a differential GPS and
stored along with the SBP data.

During processing, the SBP acoustic traces were read in using code provided by Kozola (2020) and visualized using a Matlab
script available with this manuscript. The average value of the acoustic wave velocity of the water column (1486.6 m/s for
both lakes) were used to convert the two-way travel time of the acoustic pulse into a depth scale for the seismic profiles.

## 3.3 Transient electromagnetic (TEM) soundings

Transient electromagnetic (TEM) soundings were carried out from the water surface using a single-loop configuration in March 2018. The loop with a diameter of 22.9 m (surface area: ~412 m²) consisted of a single, insulated copper wire attached to a floating ring made of twenty-four 1-inch PVC tubes. The ring was towed by an inflatable boat equipped with an electric motor, which was only turned on for navigation between sounding sites. During the measurements, the loop was separated by 5 m from the inflatable boat. Depending on the specific wind conditions, the unanchored system slowly drifted during the measurements resulting in maximum displacements of approximately two times the loop diameter (i.e., ~40 m). Due to the comparably low measurement velocity (ca. 3 min per sounding) and the poor maneuverability of the experimental setup, TEM data was acquired along a limited number of irregularly distributed lines of interest (Fig. 2a).

A simple echo-sounder (Garmin Fishfinder series) was used to measure the water depth at each sounding site. A TEM-FAST48 (manufactured by Applied Electromagnetic Research) was used for the acquisition of TEM sounding data. Transients were recorded using a transmitter current of 1 A and 32 time gates between 3.6 µs and 1024 µs after current shut-off. For this transient length, the measuring device records 64 transients, which are analogously averaged by the hardware. For one sounding measurement, this basic measuring cycle is repeated $n \times 13$ times. For $n = 4$, which we used for our measurements, this results in 52 repetitions of the basic cycle and a total of 3328 effective stacks, which are used to compute the impulse response by digital averaging and to determine the measurement error as the standard error of the mean (SEM). For times around 200 µs (the latest time gates used for the inversion), the SEM is $\lesssim 5 \cdot 10^{-9}$ V/Am². For exemplary TEM data and errors, see Figure A1 of the appendix.

During the processing, all transients were truncated to times from 21.4 and 174.5 µs and inverted using the software ZondTEM1d (A. Kaminsky, personal communication). A conventional 1D smoothness-constrained modelling approach was used to obtain a one-dimensional multilayer model (20 layers) for each sounding position separately. ZondTEM1d supports arbitrarily shaped loops, whose vertices can be defined independently for transmitter and receiver to ensure the correct interpretation of the coincident-loop data. The same software was also used to adjust layered models (5 layers). In both cases (smooth and layered model), the water depth measured with the echo-sounder was used as a-priori information by fixing the thickness of the first layer to this value. The electrical resistivity of the water layer was fixed to 25 Ωm. This value was manually adjusted to provide a good overall fit for all soundings. Especially for sites with shallow water depths and a resistive lake bed (i.e., bedrock not covered by conductive sediments), constraining the resistivity of the water layer significantly improved the imaging results. Multidimensional effects, as investigated in detail by Mollidor et al. (2013) for TEM data from a lake with steep bathymetric slopes, were not considered in the interpretation as the bathymetric variation along our survey lines was relatively gentle. Following the approach by Yogeshwar et al. (2020), which is based on the one by Spies (1989), we estimate the depth of investigation (DOI) of our TEM soundings based on transmitter area and current, average subsurface resistivity (of the smooth models), and late-time induced voltage.

## 3.4 Time-domain induced polarization (TDIP)

Time-domain induced polarization (TDIP) data was acquired with a SyscalPro Switch 48 device (IRIS Instruments) using 48 stainless-steel electrodes separated by 5 m or 10 m depending on the target. The soft and wet mud on the exposed lakebed provided a good contact between electrodes and ground. Where TDIP profiles crossed limestone outcrops, electrodes were inserted into sediment-filled fractures in order to keep contact resistances as low as possible. Measurements were carried out with injection currents between 0.5 A and 1 A, one single stack and a 50% duty cycle with 500 ms pulse length (i.e., duration

of off time is also 500 ms). After an initial delay of 20 ms after current shut off, the voltage decay was sampled in 20 time windows with a constant length of 20 ms. We used a dipole-dipole configuration combining short dipole lengths of one electrode spacing for superficial measurements with longer dipole lengths of two and four times the electrode spacing for moderate and large depths, respectively. To prevent loss of data quality due to remnant electrode polarization (e.g., Dahlin et al., 2002), the measurement protocol avoids potential readings using electrodes that had been used as current electrodes before

(Flores Orozco et al., 2012; 2018a). TDIP lines of varying length were laid out along (and parallel to) selected 2018 SBP and TEM lines on both lakes (Fig. 2a, b). In order to cover the full length of the north-south running SBP line L4 NS, TDIP lines MET19-1 and MET19-2 were carried out as a roll-along profile with an electrode spacing of 10 m and an overlap of 12 electrodes.

During the processing, we removed erroneous measurements defined as those associated with negative apparent resistivity

and/or integral chargeability readings (Flores Orozco et al., 2018b). After the removal of erroneous measurements, raw-data pseudo sections were inspected and additional outliers were defined as those readings with integral chargeability values above 8 mV/V. Based on an exemplary data set, this processing approach is further discussed the appendix. Integral chargeability values were then linearly converted to frequency-domain phase shifts assuming a constant phase angle response (i.e., no frequency dependence) following the approach outlined by Van Voorhis et al. (1973) and implemented by Kemna et al. (1999).

Finally, 2D complex-resistivity sections were reconstructed from the filtered data using the smoothness-constrained least-squares algorithm CRTomo (Kemna, 2000). 2D sections are only visualized down to an estimated depth of investigation by blanking model cells with cumulated sensitivity values two orders of magnitude smaller than the maximum cumulated sensitivity (i.e., the sum of absolute, data-error weighted, sensitivities of all considered measurements; e.g., Weigand et al., 2017).

## 215    3.5 Seismic refraction tomography (SRT)

Seismic-refraction tomography (SRT) data was acquired with the 24-channel seismograph Geode (Geometrics) and twenty-four 28-Hz geophones installed along a line at 5 m spacing in October 2019. To generate the seismic signal, a 7.5 kg sledgehammer hitting a steel plate was used at 25 shot points between the geophone positions as well as at distances of 2.5 m from the first and last geophone, respectively. At each shot point, five shots were stacked to improve the signal-to-noise ratio.

Due to the limited length (115 m between the first and the last geophone) and investigation depth, SRT data was only collected in the central parts of selected TDIP profiles (Fig. 2a, b).

During the processing, we applied a 120 Hz low pass filter on the seismic traces to remove high frequency noise and allow for a more accurate picking of first break travel times. A tomographic inversion scheme then determines the two-dimensional velocity structure below the SRT profile based on the first-arrival travel times (e.g., White, 1989). For the filtering of the

seismic traces and picking of the first arrivals, we used a Python toolbox developed at the TU-Wien. The observed travel times were inverted with the pyGIMLi framework (Rücker et al., 2017) following a smoothness-constrained scheme. Based on the ray paths computed for the resolved velocity model (e.g., Ronczka et al., 2017), we also determine the so-called ray coverage, which permits to illustrate the depth of investigation by blanking models cells that are not covered by any ray.

## 4 Results and interpretation

### 4.1 Laboratory measurements – electrical properties of sediment and water samples

The complex-resistivity measurements on the six sediment samples carried out in the laboratory (Fig. 3a) show that most resistivity values vary within a relatively narrow range between 10 and 15 $\Omega$m. Only the resistivity of one sample (TSI19-A) from the river delta in Lake Tzibaná reached values between 18 and 20 $\Omega$m. Fig. 3b shows that phase values (here $-\varphi$) in the frequency range from 1 to 10 Hz, which is mainly tested by our TDIP measurements, are roughly comprised between 0.5 and

4 mrad. Again, the only exception is sample TSI19-A with phase values of up to 6 mrad in the relevant frequency range. We attribute the atypical behavior of the sample TSI19-A to its fluvial nature (coarse grains and high organic content), while the remaining samples are clearly lacustrine (fine grains and lower organic content). The elevated phase values at high (>1 kHz) frequencies, which can be observed for all six samples, are typical electromagnetic coupling effects (Pelton et al., 1978), but do not affect our TDIP measurements, due to the long initial delay before the sampling of the voltage decay starts.

The resistivity of the two water samples from the remaining water bodies used to improve the readings of two dry samples (MET19-A and TSI19-A) were 11.9 $\Omega$m (Metzabok) and 26.8 $\Omega$m (Tzibaná), respectively. They are significantly lower than the average water resistivity of ~34.5 $\Omega$m reported by Rubio Sandoval (2019) for water samples collected from Lake Metzabok during high lake-level stands in 2016. This reduction of electrical resistivity (i.e., increase of conductivity) is probably due to the larger effect of evaporation on the salinity of small (and shallow) water bodies. Indeed, the remaining water body in Lake

Metzabok was much smaller (~50 m²) than the one in Lake Tzibaná (~5000 m²). However, a comprehensive understanding of the strong variation of water conductivity with respect to both sampling location and time is subject of ongoing limnological research in the study area.

The average resistivity of the sediment samples for the frequency range between 1 and 117 Hz (and excluding sample TSI19-A) is 12.25 $\Omega$m, which is typical for saturated clayey sediments (e.g., Reynolds, 2011). Note that due to the contribution of

surface conduction along the charged clay-mineral surfaces (Waxman and Smits, 1968), the bulk resistivity of the sediments is even lower than the average resistivity of the water (25-35 $\Omega$m during high lake-level stands). In our case, this resistivity

contrast between lake water and sediments by a factor of 2 to 3 is of particular relevance as it allows us, in principle, to detect the two materials as separate units. To our best knowledge, this is the first time that the phase spectra of fresh lake-bed sediments have been measured in the laboratory.

## 4.2 Field measurements on Lake Metzabok

In October 2019, Lake Metzabok (average depth 15 m) was completely dry, except for some residual ponds. Its sediment-covered bottom is mostly flat with steep walls (>50% slope) and some cliffs along the shore line (Fig. 4a). Only some drainage channels, steep limestone hillocks, and small ponds (Fig. 4a–d) eventually disrupt the smooth lake-bottom topography.

### 4.2.1 Profile 1 – SBP and TDIP results reveal three distinct geological units

The north-south oriented SBP line L4 NS on Profile 1 crosses various of these limestone hollocks and depressions, which are well resolved by the first reflector in the seismogram (Fig. 4e). Within the depressions between the limestone outcrops, a second reflector can be resolved, the geometry of which shows a certain consistency with the surface of the limestone outcrops. This reflector can be interpreted as the lower limit of the sediment cover. The SBP data shows that not only the elevations (outcrops) but also the depressions in the sediment cover are influenced by the topography of the underlying limestone: Both depressions, the drainage channel in the northern as well as the small pond in the southern part, are associated with local risings of the limestone surface. Based on the SBP images, the sediment thickness mostly varies between 5 and 7 m along Profile 1. Below the surface of the limestone outcrops and the lower limit of the sediment cover, respectively, we observe zones of diffuse reflectivity. These might be related to the heavily fractured and dissolved limestone. Particularly in the flat areas, the sediment cover might also be underlain by blocks of collapsed limestone with sediment filling the spaces between blocks and debris. The lakes of the study area show all characteristics of karst lakes, which are expected to originate from collapsed karst cavities, and the collapse debris should still be present below the sediment cover.

The electrical images obtained from the co-located TDIP line (MET19-1 and 2) support this interpretation: The resistivity image (Fig. 4f) shows a gross separation into two main units: The (i) sediments as well as the supposed limestone debris-sediment mixture stand out with low resistivity values between 10 and 20 Ωm, while the (ii) limestone outcrops and the deep part of the section are characterized by higher resistivity values of up to 300 Ωm. The phase image (Fig. 4g) also shows a separation into units with low and intermediate phase values. Here, a much thinner top layer (compared to the conducting layer in Fig. 4f) stands out with phase values between 0 and 5 mrad, while the limestone bedrock shows phase values >5 mrad. The capability to separate the pure sediments from the limestone-sediment mixture underlines the benefit of evaluating the TDIP phase. Due to the relatively low data cover after outlier-removal for large dipole separations, we do not interpret the phase values at depths >50 m.

For the sediment infill, both the resistivity values of 10-20 Ωm and the phase values below 5 mrad are in agreement with our laboratory measurements on the sediment samples of Lake Metzabok, corresponding to an average resistivity of ~12 Ωm and phase values (here $-\varphi$) < 4 mrad. In contrast, resistivity and phase values associated with the limestone bedrock are significantly

higher than those of the fine-grained sediment cover. The intermediate layer, which we interpret as mixture of fine-grained

sediments and the collapse debris, seems to inherit the low resistivity of the supposed clay-rich matrix, while the phase or polarization response is increased by the limestone debris.

### 4.2.2 Profile 2 – SRT measurements confirm the presence of three geological units identified in the TDIP images

The north-south oriented Profile 2 runs parallel to the last part of Profile 1 but shifted ~10 m East. It is centered at the small pond (Fig. 4d) and has a smaller electrode spacing (5 m instead of 10 m) to better resolve the sediment-limestone contact below

the bottom of the pond. The electrical images (Fig. 5a, b) show the same characteristics as seen in the corresponding part of Profile 1. Due to the higher resolution, here, we observe an internal layering of the shallow conductive units with a less conductive (30-50 $\Omega$m) top layer of ~10 m thickness and a more conductive (<20 $\Omega$m) layer that extends down to 30 m in the northern and southern parts of the line. The separation into two units becomes more obvious in the phase image, where the superficial layer is less polarizable (well below 4 mrad) than the deeper part. A few meters East of the center of the profile, the

resistive limestone bedrock crops out, which might explain the significantly increased resistivity (>100 $\Omega$m) and phase values (>6 mrad) over the first 20 m of depth below this part of the profile.

The p-wave velocity structure in the SRT image (Fig. 5c) confirms the presence of these three units: The shallowest layer, corresponding with the sediment infill characterized by velocities between 200 and 1000 m/s, which is in agreement with values for unconsolidated fine-grained sediments reported in the literature (Uyanık, 2011). In the second layer, velocities

increase to 1500-2000 m/s, and at depths between 15 and 20 m, a sudden increase to values >2500 m/s is observed. The p-wave velocities of the deepest unit agree with the lower limit of typical ranges for limestone (Reynolds, 2011), which can be explained by the high degree of fracturing and dissolution of the karst bedrock. The seismic velocities of the intermediate layer do not provide any additional information on its nature, but could well be explained by limestone debris or heavily fractured and dissolved limestone with sediment-filled open spaces.

### 4.2.3 Profile 3 – Comparison of SBP and TDIP corroborates low phase response of lake sediments

The comparison of the SBP line along the west-east directed Profile 3 with the corresponding electrical resistivity images (Fig. 6) confirms the interpretation of the electrical images: The step in the lower limit of the sediment layer around 490 m along the SBP profile, is also reflected in the resistivity structure and it is clearly resolved in the phase image. Again, the conductive unit extends far below the SBP reflector, in particular between 440-490 m along the profile. We interpret this reflector as the

contact between pure sediments and the mixed sediment-collapse debris. Thus, the mixed layer has a lower resistivity than the superficial fine-grained sediment layer. Along this profile, both sediment-bearing layers are characterized by low phase values. The sediment-covered limestone bedrock between 440 and 530 m is characterized by high phase values, while phase values decrease as this unit approaches the surface and crops out at the end of the profile. The low phase values of the limestone outcrop do not fit the previously stated general characteristics of this unit but might be related to variations in composition

and/or degree of fracturing of the limestone bedrock.

### 4.2.4 Profile 4 – Water-borne TEM and terrestrial TDIP measurements reveal consistent resistivity models

Fig. 7a shows the electrical resistivity image reconstructed from 12 TEM soundings along the profile crossing Lake Metzabok from West to East. Both, smooth and layered models, recover a conductive layer of varying thickness below the lake floor indicating the presence of fine-grained sediment infill across the entire basin. This layer only disappears close to the shoreline

(i.e., towards the eastern end of the profile), where the resistive limestone bedrock is in direct contact with the water body. According to the layered model, the resistive bedrock itself is encountered at depths of approx. 15-20 m below the lake bed and only disappears below sounding MET10, where a possible fracture zone might be responsible for a lower resistivity at depth.

At both ends of the profile, and in particular at stations MET1 and MET2, a conductor is indicated below the resistive bedrock,

which could point to a more fractured bedrock or a lithological contact, e.g., with a shaly geological unit. However, in absence of complementary information, such as a detailed geological map or bore hole data, we can also not discard artefacts due to distorted late-time transient data. Especially close to the shoreline, where the lake bottom rises steeply, the TEM transients might be affected by multidimensional effects (e.g., Mollidor et al., 2013), which are not taken into account by the chosen one-dimensional inverse modelling approach.

Due to the relatively high average resistivity of the subsurface along this profile, the depth of investigation (DOI) computed after Spies (1989) and Yogeshwar et al. (2020) is mostly larger than the 80 m shown here. Possibly due to the large resistivity and thickness of the limestone bedrock, no changes of the modelled resistivity have been observed at depths >80 m.

Between stations MET3 and MET7, the TEM image recovers a resistivity distribution similar to the one of co-located TDIP profile (Fig. 7b). Taking into account that the water-borne TEM survey was carried out with an average of 15 m water column,

the consistency with the TDIP resistivity results from the lake bed clearly indicates a good quality and reliability of the obtained TEM imaging results.

As observed before, the phase image (Fig. 7c) shows a non-polarizable top layer, which at a depth of ~10 m is underlain by a unit with a higher polarization response (absolute phase values around 10 mrad and higher), corresponding with the debris-sediment unit. The SRT tomogram (Fig. 7d) shows a sharp increase in p-wave velocity at depths between 20 m (in the western

part) and 30 m (in the eastern part). This southeast-dipping surface correlates with a similar structure in the TDIP resistivity model, which we again interpret as the contact with the limestone bedrock.

### 4.2.5 Geological interpretation of the geophysical survey on Lake Metzabok

The schematic sketch presented in Fig. 8a summarizes our geological interpretation of the geophysical profiles of Lake Metzabok and the observations made on the exposed bed of the drained lake (Fig 4 a-d and Fig. 8b-d). The model rests on the

assumption that the lakes in the study area are formed by the coalescence of a number of dolines that resulted from the collapse of karst cavities. The remains of the collapsed limestone are expected to have formed a debris layer covering the floor of the former caves. Subsequently, the fluvial input of fine-grained lake sediments has first filled up the interspaces between the

blocks and then buried the collapse remains, forming the two uppermost units observed below all profiles. Fig. 8b and c show pictures of such mixed materials exposed on the surface of the drained Lake Metzabok.

Table 1 summarises the physical properties of the main units of this geological interpretation. The electrical resistivity of the fine-grained sediments and the mixed collapse debris and sediment layer is comprised within a relatively narrow range. In the TEM and TDIP resistivity images, these two units may appear as one conductive unit (see red dotted lines in Fig. 8a). It is not clear, why the addition of the more resistive lime stone debris should decrease the resistivity of the mixed unit compared to the pure fine-grained sediments. In terms of the phase values, the distinction between these units is clearer and the increase of

the phase response in the mixed layer is straightforward (because the limestone is more polarizable than the fine-grained sediments based on our field measurements). The clearest indication of the inner structure of the conductive unit comes from the collocated SBP sections, which show a clear seismic reflector at the corresponding depth. The limestone bedrock becomes detectable by its high p-wave velocity in the SRT images and its high resistivity (TEM and TDIP), while its phase response varies over a larger range and is not as unambiguous. It is worth mentioning that wherever the fine-grained sediments are

underlain by the collapse debris layer, the limestone bedrock does not appear as an additional reflector in the SBP sections.

## 4.3 Field measurements on Lake Tzibaná

While the 2019 lake-level decrease left Lake Metzabok (max. depth 25 m) completely drained, the deeper Lake Tzibaná (max. depth 70 m) always preserved a water cover on at least 2/3 of its maximum surface area. The long N-S oriented SBP section in Figure 9 crossing the entire Lake Tzibaná (Profile 5) shows a similar lake-bottom architecture as the one derived for Lake

Metzabok: Steep limestone walls along the shoreline and flat parts with the typical 3-layer structure consisting of fine-grained sediment cover, collapse debris, and limestone bedrock. Unlike in the case of Lake Metzabok, the flat parts of Lake Tzibaná are found at two different levels, which are separated from one another by a steep limestone cliff. Additionally, the southern part of the profile crosses the delta of the Nahá river, where we expect a higher fraction of coarser material (sand/gravel) in the fluvial deposits in comparison to the well sorted sediments, mainly composed of clay and silt, covering the flat parts of the

lake bottom. In the SBP profile, these delta deposits stand out by a highly reflective lake bottom, which results in strong multiple reflections between lake bottom and water surface. Yet, such reflections inhibit the recovery of any information on the internal structure of the delta.

The TEM, TDIP, and SRT measurements carried out along Profile 6, which roughly covers the last 450 m of Profile 6 (see survey layout in Fig. 2b) fill in this missing information. The resistivity images of both TEM and TDIP measurements presented

in Fig. 10a, b consistently show three main units: (1) the resistive (>100 Ωm) limestone bedrock at depth, (2) a highly conductive (<10 Ωm) clay layer on top, and (3) a layer of intermediate resistivity (~30-100 Ωm), in particular between 200 and 400 m, corresponding to the sand banks and possible interbedded strata of clay, sand, and gravel associated with the delta deposits. As observed above along Profile 4, the resistivity model of TEM sounding TZI41 indicates a conducting unit below the resistive limestone bedrock, which could be related to a lithological contact, a fracture zone, distorted late-time data, or

multidimensional effects. The low average resistivity below soundings TZI13-44 result in a significantly reduced depth of

investigation (approx. 55-60 m). The lack of borehole data hinders a conclusive interpretation of this conductive anomaly at depth.

Probably due to the highly heterogeneous composition of the river delta, these deposits also show a heterogeneous distribution of phase values (Fig. 10c). As observed before, the clay and limestone units below the fluvial deposits show low and high phase values, respectively. The relatively high phase values in the clay layer below the fluvial deposits are probably inversion artefacts caused by the relatively noisy TDIP data along this line.

The SRT image (Fig. 10d) shows p-wave velocities as low as 100-200 m/s within the fluvial deposits, which are in agreement with literature values for partially saturated, unconsolidated sand (e.g., Barrière et al., 2012). P-wave velocities increase with depth across the thick (and probably compacted) clay layer. According to the electrical images, the surface of the bedrock lies below the lower limit of the SRT image. Accordingly, the highest velocities of <2000 m/s, seen in the SRT image, do not reach the high values (>2500 m/s) typical for limestone bedrock.

## 5. Discussion

### 5.1 Identification of suitable drilling locations

Our geophysical investigations delineate a 5-6 m thick and nearly undisturbed layer of fine-grained lacustrine sediments covering the flat parts of Lake Metzabok. Such a layer is relevant for the conduction of paleolimnological perforations. Suitable drilling locations can be defined between 450 and 550 m, as well as between 600 and 700 m along Profile 1 (SBP profile in Fig. 4e). The large variation of the sediment thickness observed along Profile 3, which is perpendicular to Profile 1, underlines the need for a comprehensive pre-drilling investigation and an accurate positioning of the drilling equipment. The sediment layer between 100 and 200 m along Profile 5 represents a suitable drilling location for Lake Tzibaná (sediment thickness also 5-6 m). Although the deeper part of Lake Tzibaná is covered by sediments (between 400 and 600 m along Profile 5), too, sediment thicknesses in this part of the lake are smaller (only 3-4 m according to the SBP image) and drilling efforts would be considerably higher, due to the larger water column at this location (approx. 30 m during water-level high stand in 2019). With a thickness of >40 m according to our electrical imaging results, the sedimentary cover along Profile 5 of Lake Tzibaná (particularly between 250 and 400 m) is much thicker than the sediments covering the flat parts of both lakes. However, our results also indicate that these sediments rather correspond to fluvial deposits of the river delta. In this depositional regime, we expect much higher rates of sedimentation and thus not necessarily an older paleoenvironmental record. Additionally, river deltas are much more dynamic systems, in which sediments are deposited, eroded, and redeposited repeatedly, which decreases the probability to obtain undistorted sediment records as encountered farther offshore.

### 5.2 Geological situation of the studied lakes and hydrogeological implications

Our field observations and geophysical imaging results also have important implications for the general understanding of the geological situation of the two studied karst lakes: Large areas of both lakes are covered by a layer of clayey sediments, which

have a low hydraulic permeability. Thus, where this layer is thick enough (up to 5-6 m across large areas), it acts as a hydrological barrier between the lakes and the underlying karst. However, the remaining heavily fractured and uncovered limestone outcrops (e.g., Fig. 8b-d) effectively connect the lakes with the karst water system. This conclusion is underscored

by the high velocity at which the two lakes drained practically simultaneously between February and July 2019. Accordingly, the sudden drainage of both lakes might be related to the same hydrogeological process.

While the interconnectivity between surface water and karst aquifer is well documented by field observations and further supported by the interpretation of our geophysical results (see Fig. 8), the specific cause(s) and mechanism(s) of the sudden drainage of Lakes Metzabok and Tzibaná remain unrevealed. The suddenness of the drainage suggests that one or more

previously clogged karst conduits were unplugged around these dates. Planned time-series analyses of hydrological and meteorological data in combination with paleoenvironmental studies on sediment cores will possibly provide more detailed insight into the mechanism and its triggers, and thus shed light on the question whether such catastrophic drainage events as the one observed during 2019 are linked to recent climate change or another geodynamic process.

## 5.3 Lessons learned from implementing a multi-methodological approach for lake-bottom reconnaissance

Only the combination of complementary methods employed in the present study allowed us to produce comprehensive geological models of the lake-bottom geology of the studied karst lakes. Table 2 summarizes the characteristics of the four field methods (SRT, TEM, TDIP, and SRT), the individual contributions of each method and their respective limitations identified in this study. In the following, we will discuss some important aspects of this overview in more detail.

### 5.3.1 Sub-bottom profiler reflection seismic method

For shallow-water applications, the compact and mobile experimental setup of the SBP technique offers clear advantages. Additionally, the high productivity and resolution in combination with the straight-forward interpretation of the SBP seismograms evidence that such reflection seismic methods are best suited for a first reconnaissance of the lake bottom. In comparison, water-borne TEM measurements are by far slower and more labour intensive (for both data collection and processing) and the resulting imaging results have a lower lateral resolution. In our study, the contact between fine-grained

clay sediments and the underlying mixed layer (collapse debris and sediment) was clearly visible from the SBP data, which permits a straight-forward estimation of sediment thicknesses along SBP survey lines. The contact or transition between mixed layer and limestone bedrock was also noticeable in the SBP images but the interpretation was not as clear as in the case of the first two layers and mainly built on the availability of complementary TEM and TDIP data. The main limitations of the SBP survey consist in the low depth of penetration of this method, which hardly reached 10 m below the lake bottom, and in the

total lack of sub-bottom information as soon as the lakebed is covered by coarser sediments as observed in the deltaic region of Lake Tzibaná. Such "opaque seismic facies and high […] reflectivity" of fluvial sediments have been discussed before by Orlando (2013) for measurements on the river Tiber in central Italy. Hence, the combination of waterborne TEM and SBP methods could offer a solution to improve the investigation of deep areas (resolved by TEM data), while lateral information

can still be gained using SBP. The inclusion of SBP information for the interpretation of TEM data towards the inversion of an improved resistivity model, is an open area of research.

### 5.3.2 Water-borne transient electromagnetic method

The water-borne TEM sounding system developed for this study turned out to provide reliable resistivity images for water depths down to at least 20 m. This conclusion is supported by the agreement of the resistivity images obtained from water-borne TEM and lake-floor TDIP measurements along both TEM profiles (Fig. 7 and Fig. 10). Previous shallow-water TEM studies (e.g., Butler, 2009; and references therein; Hatch et al., 2010; Mollidor et al., 2013) employed in-loop configurations with an outer transmitter loop and a smaller receiver loop or coil in the centre, while we used a light-weight single-loop configuration, which is quicker to assemble and easier to handle while navigating on the lake. It is worth mentioning that in terms of noise level and depth of investigation, our simple system consisting of one single circular loop provides comparable results as those obtained with more sophisticated systems consisting of separated transmitter and receiver square loops (e.g., Yogeshwar et al., 2020).

Besides the use of a single-loop configuration, the use of small loops as employed in the present study can eventually result in distortions in the transient data. The measured curves (truncated to 21.4–174.5 μs) do not show any conspicuous features (see data example in the appendix) and can be adjusted by reasonable resistivity models with an overall low root-mean-square (RMS) deviation. Thus, we discard the presence of adverse effects in our data set. The good agreement between TEM and TDIP-derived resistivity models along collocated survey lines further supports this conclusion.

In the present study, TEM resistivity images clearly delineate the top of the bedrock and reveal the inner structure of the deltaic deposits of the river Nahá, which are not resolved by the SBP seismograms. The interpretation of the layered resistivity structure below the flat parts of the lake bottom is only possible by combining TEM resistivity images with complementary information from other methods. In particular, the SBP seismograms (and TDIP phase images) imply that the thickness of the fine-grained lakebed sediments does not exceed 5-7 m in Lake Metzabok, while conductive units extend down to depths of 20-30 m (below the lake bottom) and more. We resolve this apparent contradiction by postulating an intermediate layer made of limestone debris from collapsed karst cavities and fine-grained sediments filling the spaces between the limestone blocks. This mixed layer seems to be characterized by a much higher seismic velocity compared to the fine sediments but a similar or slightly lower electrical resistivity. Consequently, the small resistivity contrast between the fine-grained lakebed sediments and the underlying mixed layer hinders an unambiguous estimation of sediment thickness from TEM (and TDIP) resistivity data alone.

Our results also show the advantages in the interpretation of the sounding data after the incorporation of water depth and eventually water resistivity into the inversion of TEM data as a-priori information. Water depth is readily measured during the TEM sounding using a standard echo sounder. Further investigations can consider the addition of fluid conductivity and temperature measurements using conductivity-temperature-depth (CTD) probes to improve the inversion of waterborne

measurements and, thus, the investigations of the lakebed by electrical methods. Such information can also be obtained from the analysis of water samples.

We have adjusted smooth and layered models to the TEM sounding data, both recovering similar sub-bottom structures along
the two lines discussed here. While the smooth models facilitate a direct comparison with the (smooth) 2D TDIP resistivity images, the layered models are more appropriate to locate sharp geological contacts. The average fit quality (as assessed by the percentage RMS), which is slightly better for the layered models, could point to rather sharp contacts. However, it is not at all straight forward to decide whether sharp resistivity contrasts exist between the main lithological units (i.e., sediment cover and limestone) or not. As our interpretation of the Metzabok data suggests, e.g., within the mixed layer, there might be
a smooth transition due to a continuously increasing volume content of limestone with depth. The same is true for contacts between different, eventually interbedded sedimentary units (e.g., fine-grained lake sediments/sandy delta deposits).

### 5.3.3 Induced-polarization imaging of the lake floor

The fact that the studied lakes drained provided us with the unique opportunity to carry out TDIP measurements directly on the lake floor. The low contact resistances and the easy installation of electrodes on the soft ground represent ideal conditions
for electrical imaging measurements. Furthermore, the evaluation of both phase data for sediment samples analysed in the laboratory and for the in-situ measurements on the exposed lake floor is unprecedented or at least very rare in geophysical literature. In the present study, the TDIP phase results permitted to significantly improve the obtained geological model of the lake-bottom. In particular, the IP images showed a low phase response of the lake sediments on one hand and a comparably high phase response of the limestone bedrock and the collapse-debris layer on the other hand. The interpretation of the field
TDIP phases is sustained by our laboratory measurements on sediment samples and the good overall agreement of the shallow low-phase layer with the corresponding reflector in the SBP images.

Larger variations in the phase response of sediments (especially the increased phase values of sample TZI19-A) are likely related to different depositional regimes: Preliminary geochemical analyses of the sediment samples imply significantly increased total organic carbon (TOC) and carbon-to-nitrogen ratio (C/N) of the sample TSI19-A compared to the other five
samples (P. Hoelzmann, personal communication). High values of TOC and C/N point to a larger fraction of organic matter from terrestrial sources in sample TZI19-A, while the smaller amount of organic matter of the other five samples probably stems from algal plants. A strong control of TOC on the phase response has been reported earlier for other materials (e.g., Schwartz and Furman, 2015; Flores Orozco et al., 2020).

Based on the encouraging findings of the present study, the application of TDIP imaging for the lake-bottom characterization
emerges as an interesting complementary method for the characterization of lake-bottom sediments. Although promising for desiccated or shallow lakes, there are some limitations for TDIP measurements to be carried out on water-filled lakes: In principle, TDIP data could be collected with floating electrode arrays as used for water-borne direct-current resistivity surveys. However, the collection of deep IP data - as needed to investigate the sediments below a water column of, e.g., 20 m – often

suffers from a low S/N ratio. This limitation could be overcome by bringing the electrodes closer to the lake bottom, which is possible but logistically more effortful (e.g., Baumgartner and Christensen, 2006).

### 5.3.4 Seismic refraction tomography of the lake floor

In the present study, the land SRT measurements carried out on the exposed lake floor confirmed the layered structure of the flat parts of the bottom of Lake Metzabok inferred from the preceding three methods. In those cases, where the depth of investigation of the SRT images was large enough to cover the top of the limestone bedrock (i.e., Profiles 2 and 4), this geological contact was delineated clearly by a steep increase in the SRT velocity model. The main limitations regarding the applicability of SRT measurement on the lake floor, which we identified in this study, are related to the specific surface conditions: On the one hand, the generation of seismic pulses was excessively labour intensive, as the steel plate bogged down into the soft lake-bottom and had to be dug out after every single hit. On the other hand, the low signal-to-noise (S/N) ratio of some data collected on the muddy lake floor (e.g., along Profile 2, see appendix), rendered the processing and interpretation of SRT results challenging. We attribute the low S/N to the difficult coupling of seismic energy into the ground, a high energy loss of seismic signals in soft sediments, and a higher level of ambient noise (e.g., induced by wind). Although the picking percentages of noisy SRT profiles (here, Profile 2) were much lower and RMS deviations significantly increased in comparison to data collected on firm ground (e.g. Profile 5), the depth of investigation only decreased by approx. 20% (see Figs. 5c and 7d).

### 6. Conclusions

Based on the combination of different geophysical techniques, the present study provides important insight into the geological situation of two hydraulically highly dynamic karst lakes. The comparison of water-borne and land surveys (carried out after the sudden drainage of the lakes) permits a detailed evaluation of the potential and limitations of different seismic, electrical, and electromagnetic geophysical methods for the investigation of such lakes. One principal outcome of this study is that only the combination of complementary methods provides sufficient information to develop a comprehensive geological model of complex karst environments. In this sense, the present systematic field study paves the way towards an improved geophysical characterization, which is needed to better understand surface-groundwater interactions in karst systems and, more importantly, to evaluate climate-change related effects on karst water resources. In this regard, the interpretation of our results permitted to determine suitable drilling locations for future paleoenvironmental drilling campaigns, which are characterized by thick (5-6 m), undisturbed fine-grained lake sediments covering the flat parts of both studied lakes. The recovery of continuous and far-reaching sedimentary records is another important element for the understanding of the impact of climate change on the availability and quality of water in karst systems.

The possibility to recollect additional data directly on the exposed lake floor after the sudden drainage of Lakes Metzabok and Tzibaná substantially benefitted the evaluation of the different methods. The good agreement of electrical resistivity data

collected with a new water-borne TEM system, TDIP resistivity data from the dry lake bottom, and electrical measurements on sediment samples in the laboratory demonstrates that the new TEM system works well down to water depths of at least 20 m. Furthermore, there is no reason to assume that the system should not work as well in even deeper water (>20 m) depending on the water conductivity. TDIP phase images turned out to provide relevant complementary information – here, in particular about the inner structure of the conductive units covering the lake bottom. Seismic data from a water-borne SBP and a SRT

survey on the dry lake floor provided complementary information and allowed to interpret the two conductive units as a top layer of fine-grained sediments and an intermediate layer of debris from collapsed cavities in the heavily karstified limestone bedrock. At the same time, the delineation of the upper limit of the buried limestone bedrock was not possible from the seismic data alone. Thus, the final interpretation was only possible by combining electrical and seismic data sets and by incorporating geological and geomorphological constraints, showing – once again – the strength of a multi-methodological and

interdisciplinary approach.

## Appendix A: Data quality and filtering

TEM. Figure A1a shows the TEM raw data of selected soundings along Profile 6 of Lake Tzibaná in terms of the induced voltage (normalized to loop area and transmitter current). As described in the main text, all sounding curves were truncated to a unit time window between 21.4 and 174.5 μs. Earlier times were ignored to minimize the effect of distorted early-time data.

At the latest time window, the SEM of the induced voltage is approximately $5 \cdot 10^{-9}$ V/Am$^2$ for all soundings (except for TZI44) and about 1–2 orders magnitude smaller than the corresponding induced voltage. Figure A1b shows the measured and calculated apparent resistivity curves of the same soundings. As indicated by the overall small root-mean-square error (RMS<5%), the measured curves are all recovered well by the adjusted smooth resistivity models, which are visualised in Fig. 10.

TDIP. TDIP data was filtered based on the apparent resistivity and apparent chargeability data. In a first step, TDIP readings with apparent resistivity values ≤0 Ωm and/or apparent chargeability values ≤0 mV/V were removed as outliers. In a second step, based on the visual assessment of the raw data pseudo sections and histograms (see Fig. A2 for an exemplary data set), measurements with apparent chargeability values >8 mV/V were removed as further outliers. The selection of the limit of 8 mV/V for the apparent chargeability values is based on the observation of a narrow distribution of physically meaningful

values in the corresponding histograms (see Fig. A2d and h). In the case of the data set shown in Fig A2, which corresponds to the second part of the roll-along profile 1 of Lake Metzabok, this filtering results in a reduction to 57% of the unfiltered data set. This high loss of data is related to a comparably poor data quality of the chargeability measurements along this long line (470 m length, 10 m electrode spacing). Shorter profiles with half the electrode spacing (5 m in the case of profiles 2-4 and 6) are less affected by noisy chargeability data as reflected in a higher percentage of useful data (up to ~88% in the case of Profile

3 of Lake Metzabok).

SRT. Collected with 24 geophones and 25 shot positions, each tomographic data set consists of a total of 600 seismic traces. Fig. A3 shows exemplary seismic traces for one central shot positions and the travel-time curves (constructed from the picked first arrivals of all 25 shot positions) for one relatively noisy (Profile 2) and one relatively clean (Profile 4) data set. The picking percentage displayed along with the travel-time curves reflects the number of traces, for which a first arrival could be identified, and serves as a measure of overall data quality. The low data quality of Profile 2 data results in a low picking percentage (341 out of a total of 600 traces) and mainly affects long-offset data, which clearly reduces the depth of investigation (~40 m, Fig. 5c). In comparison, SRT data collected along Profile 4 is cleaner (552 first out of 600 first arrivals picked) and, thus, results in a larger depth of exploration (>50 m, Fig. 7d).

**Data availability**

All raw and processed data of this study (and some additional data not discussed here) are available at Zenodo (https://doi.org/10.5281/zenodo.3782402) along with the Matlab scripts used to prepare the visualizations presented in this manuscript.

**Author contributions**

AFO, JG, JH, WM, EG, and JR participated in the field seasons, which were planned and coordinated by MB, LP, AFO, CP, AH, and AS. RG and JH were responsible for the sediment and water samples and the laboratory IP data. CP, EG, and JR processed the SBP data. AFO processed and inverted the TDIP data, MS the SRT data, and LA the TEM data. WM made a geological field survey and gave insights into the geological context. JB prepared the maps and participated in the geological contextualization. All authors participated in the interpretation and discussion of the results. MB lead the redaction of the manuscript with contributions of all co-authors.

**Competing interests**

The authors declare that they have no conflict of interest.

**Acknowledgements**

We thank the Comisión Nacional de Áreas Naturales Protegidas (CONANP) and the authorities of the protected area Nahá and Metzabok, in particular Sergio Montes Quintero, Santiago Landois Álvarez Icaza, Miguel García Cruz, Rafael Tarano, and José Ángel Solórzano, as well as the municipalities of Nahá and Metzabok for their openness and friendly support. We are grateful for the help provided by Mauricio Bonilla, Johannes Bücker, Martín Garibay, Carlos Cruz, Roberto Reyes, Lorena Bárcena, Rodrigo Martínez Abarca, and Theresia Lauke, and all other colleagues and students, who were actively involved

during the field seasons. Finally, we would like to thank Socorro Lozano, Margarita Caballero, Beatriz Ortega, Sergio Rodríguez, and Alex Correa Metrio from the Institutes of Geology and Geophysics, UNAM, for institutional and logistical support. We also thank Pritam Yogeshwar and one anonymous reviewer for their thoughtful revisions, which helped to significantly improve the present manuscript.

**Financial support**

This project was funded by the Consejo Nacional de Ciencia y Tecnología (CONACyT) under grant number 252148 and the Deutsche Forschungsgemeinschaft (DFG, German Research Foundation) – Project-ID 439783529. Parts of this work were funded through the Austrian Science Fund (FWF) – Agence Nationale de la Recherche (ANR) research project FWF-I-2619-N29 and ANR-15-CE04-0009-01 HYDROSLIDE: Hydro-geophysical observations for an advanced understanding of clayey landslides as well as by the Austrian Federal Ministry of Science, Research and Economy (project: ExploGRAF- Development of geophysical exploration methods for the characterization of mine-tailings towards exploitation).

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

**Table 1: Ranges of physical properties of the geological units interpreted from our geophysical profiles and laboratory measurements. Resistivity data based on TEM, TDIP resistivity and laboratory measurements; phase data (absolute value)**
**according to TDIP images and laboratory data between 1 und 10 Hz; p-wave velocity from SRT images.**

| Material / geological unit | Resistivity (Ωm) | Phase (mrad) | P-wave velocity (m/s) |
|---|---|---|---|
| Fine-grained sediments | 5–30 | <4 | 200–1500 |
| Collapse debris & fine-grained sediments | 5–20 | >5–6 | 1500–2000 |
| Limestone bedrock | >100 | >4–5 | >2000 |

**Table 2: Methods used to image the sub-bottom structure of the studied lakes. Summary of physical properties resolved, typical parameter ranges in the study area as well as setup and characteristics of the measurements. Typical depths of investigations depend on the specific instrumental setup used; main contributions and limitations mostly refer to the present study and the specific geological situation.**

| Method | Physical property and typical range | Setup and characteristics | DOI (m) | Main contribution | Main limitations |
|---|---|---|---|---|---|
| SBP | $v_p$* only reflection patterns resolved | Water borne; 10-kHz transducer, 300 W output power, boat | 10 | Resolves contact between fine-grained sediments and underlying mixed layer (e.g., Fig. 4e) | – Penetration depth $\leq$ 10 m below lake bottom (e.g., Fig. 4e) <br> – No penetration in coarse delta sediments (e.g., Fig. 10a) |
| TEM | $\rho$** 5–500 Ωm | Water borne; unanchored single-loop system, 412 m² loop area, 1 A transmitter current, rubber boat | 50–100 | Delineates top of bedrock (e.g., Fig. 7a) and coarse delta deposits (e.g., Fig. 10a) | – Low acquisition velocity/productivity compared to SBP |
| TDIP | $\rho$ 5–500 Ωm <br><br> $\varphi$*** 2–6 mrad | Terrestrial; 5–10 m spacing, 48 electrodes, dipole-dipole, 0.5–1 A transmitter current, 500 ms pulse | 50–70 | $\rho$: Delineates coarse delta deposits (e.g., Fig. 10b) <br> $\varphi$: Improved delineation of fine-grained sediments (e.g., Fig. 5b) | – No clear distinction between lake sediments and limestone bedrock if only $\rho$ is considered (e.g., Fig. 4f) |
| SRT | $v_p$ 200 – 3000 m/s | Terrestrial; 5 m spacing, 24 geophones (28 Hz), energy source: 7.5 kg sledge hammer | 40–50 | Delineates top of bedrock (e.g., Fig. 7d) | – Eventually low quality of data acquired on muddy lake floor |

*p-wave velocity, **electrical resistivity, ***resistivity phase

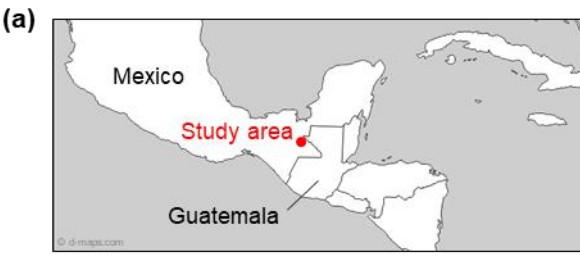

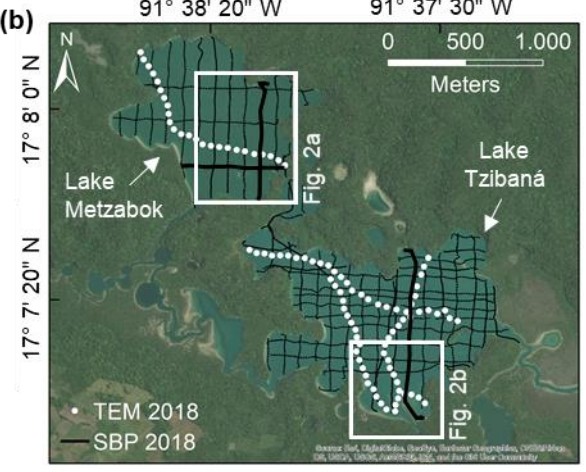


**Figure 1: (a) Location of the study area in Southern Mexico (from d-maps.com). (b) Layout of the geophysical survey on lakes Metzabok and Tzibaná during high-level stands in March 2018. Black lines show the sub-bottom profiler (SBP) survey grid, bold lines highlight those profiles discussed in detail in this manuscript, white circles represent individual transient electromagnetic soundings (TEM). The optical satellite image in the background (source: Bing Maps data base) shows lake water surface similar to**
**the high-level stands encountered during March 2018.**

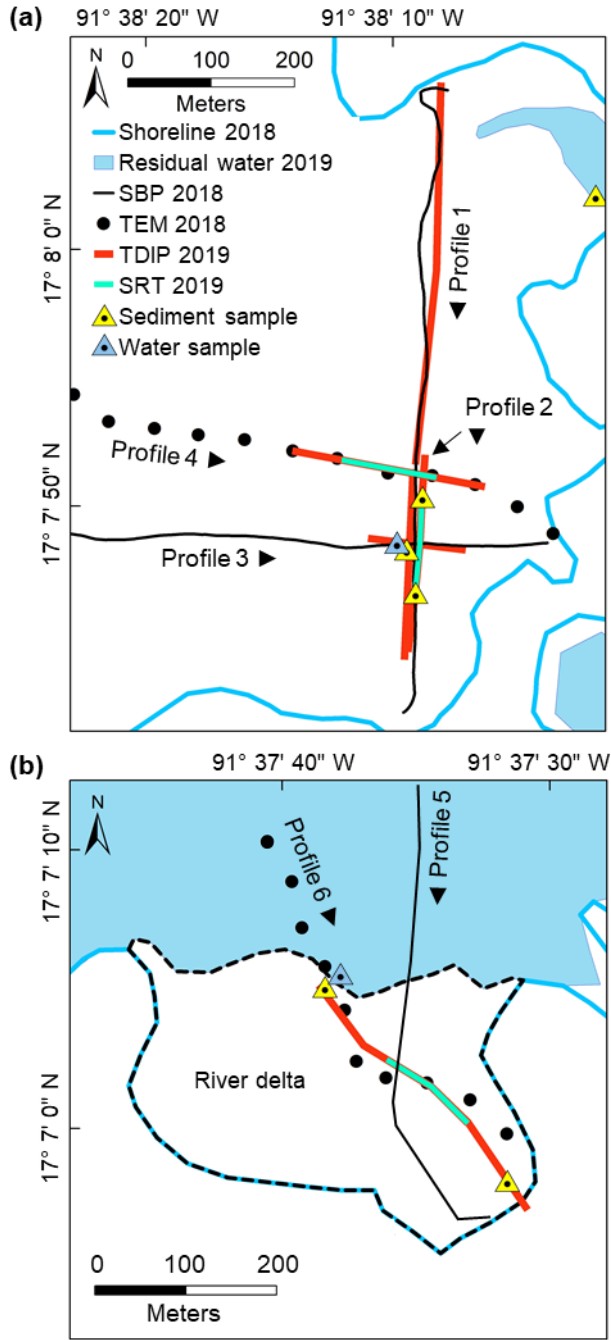

**Figure 2: Layout of the geophysical survey on lakes Metzabok (a) and Tzibaná (b) in October 2019 (after the sudden lake-level drop) including sub-bottom profiler (SBP), transient electromagnetic (TEM), time-domain induced polarization (TDIP), and seismic refraction tomography (SRT) measurements. The geophysical measurements discussed here are grouped into five profiles; black triangles next to the profile names indicate the profile orientations. Yellow and blue triangles indicate sampling locations for sediment and water samples analysed in the laboratory, respectively. The dashed black line in (b) indicates the dry part of the river delta exposed during October 2019.**


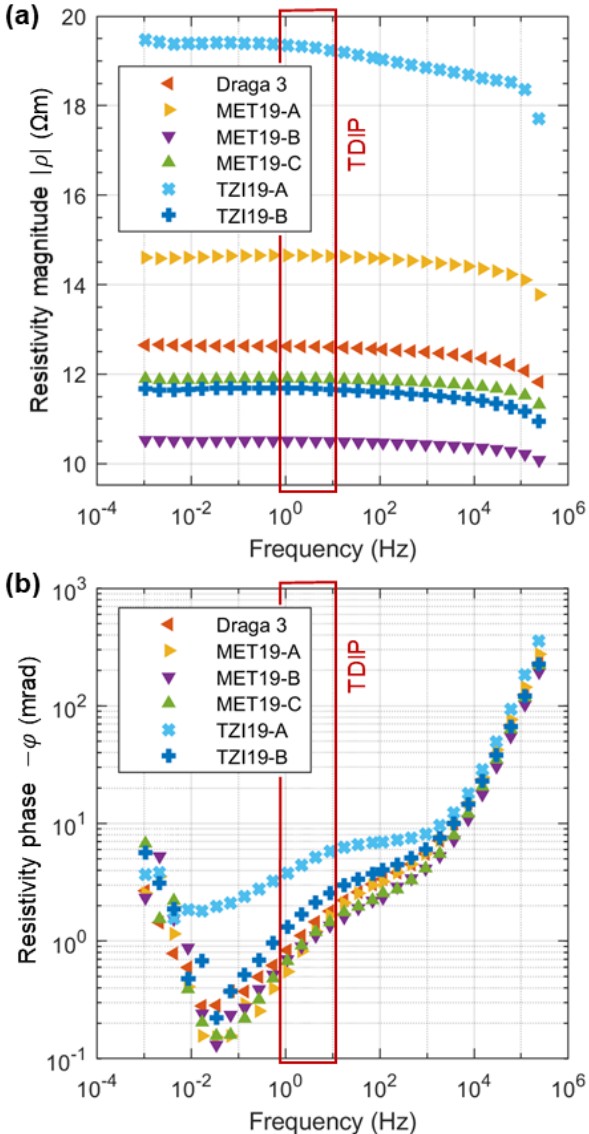

**Figure 3: Frequency-dependent complex resistivity of six lake-bottom sediment samples retrieved from Lake Metzabok (triangles) and Lake Tzibaná (crosses). Complex-resistivity values are given in terms of (a) magnitude and (b) phase. The highlighted frequencies between 1 and 10 Hz roughly correspond to the range tested by our time-domain induced polarization measurements in the field.**

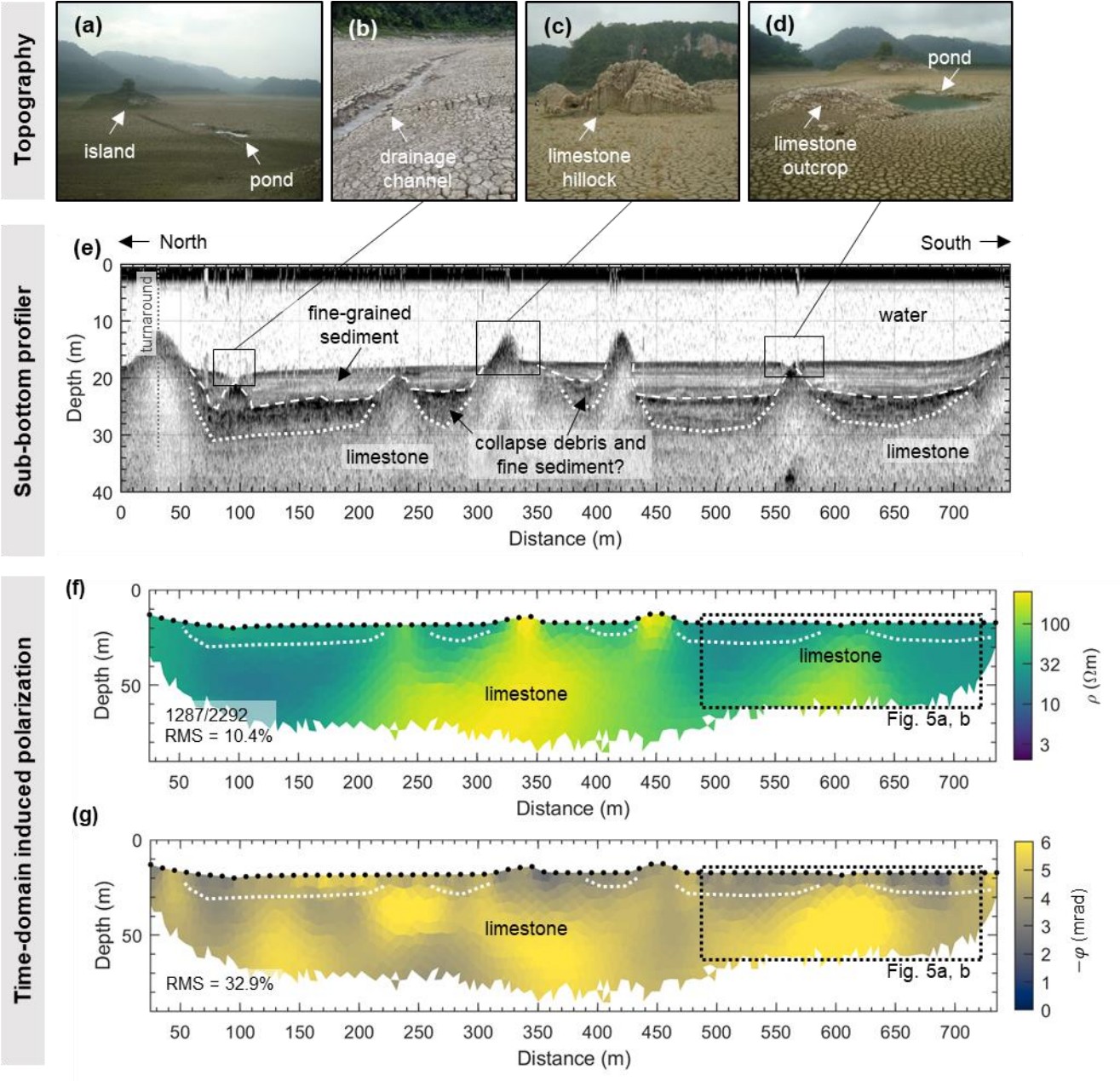

**Figure 4: Topographic features and geophysical sections along Profile 1 of Lake Metzabok. Photographs taken in October 2019: (a)**
**lake basin with flat bottom, (b) drainage channel, (c) limestone hillock, (d) deep fracture and pond next to shallow limestone outcrop. (e) Sub-bottom profiler (SBP) section with dashed lines highlighting the main reflector encountered below the lake floor and dotted lines outlining a zone of high diffuse reflectivity, (f) and (g) electrical resistivity and phase images, respectively, including electrode positions (black dots along the surface) and dotted lines taken from SBP section. Electrical sections are shifted by 25 m with respect to the SPB section. Labels in the lower left corners of (f) and (g) represent the amount of data points used for the inversion compared**
**to the total measured data (same for resistivity and phase) and the respective percentage root mean square deviations (RMS) of the inversion.**

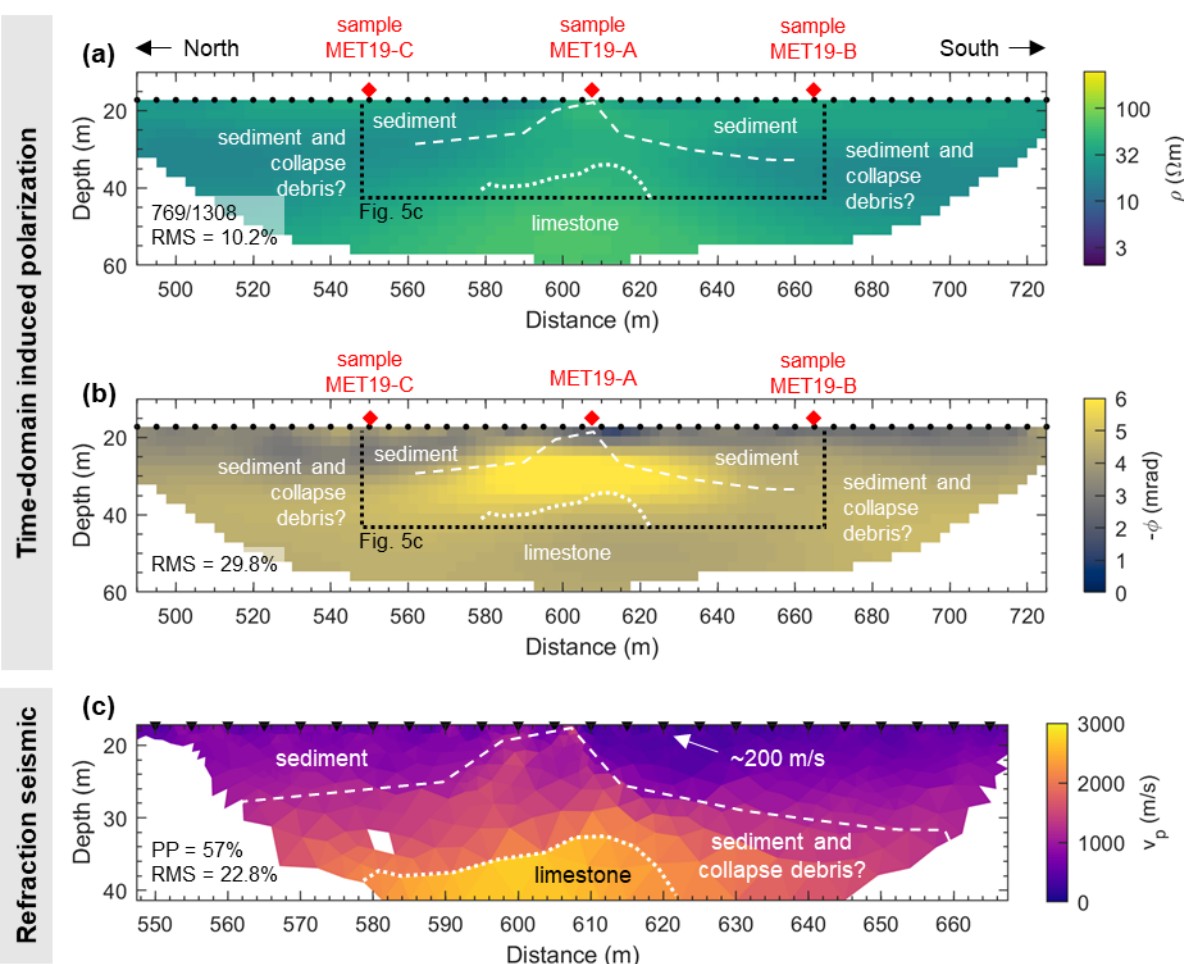

**Figure 5: Geophysical sections along Profile 2 of Lake Metzabok (same coordinates as in Fig. 4): (a) and (b) electrical resistivity and phase images, respectively, including electrode positions (black dots along the surface), sampling locations of sediment samples (red diamonds at the surface), and the main lithological units interpreted from the SBP image (dotted lines). Labels in the lower left corners represent the amount of data points used for the inversion compared to the total measured data (same for resistivity and phase) and the respective percentage root mean square deviations (RMS) of the inversion. (c) Seismic refraction tomogram with main lithological units including picking percentage (PP) and RMS of the inversion.**


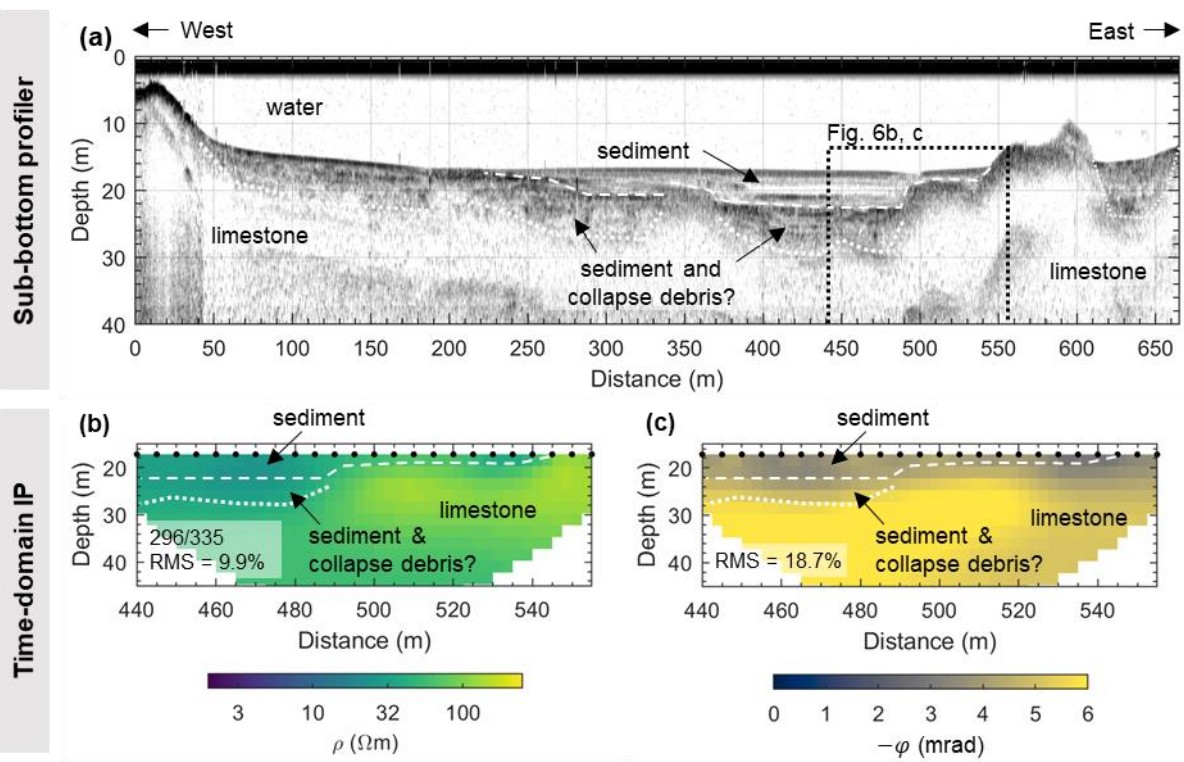

Figure 6: Geophysical sections along Profile 3 of Lake Metzabok: (a) sub-bottom profiler section with dashed lines highlighting the main reflector found below the lake floor, dotted lines outlining a zone of high diffuse reflectivity including main reflectors and the dashed box showing the section with TDIP resistivity and phase data, (b) electrical-resistivity image and (c) phase image including electrode positions (black dots along the surface) and lines taken from SBP seismogram. Labels in the lower left corners represent the amount of data points used for the inversion compared to the total measured data (same for resistivity and phase) and the respective percentage root mean square deviations (RMS) of the inversion.

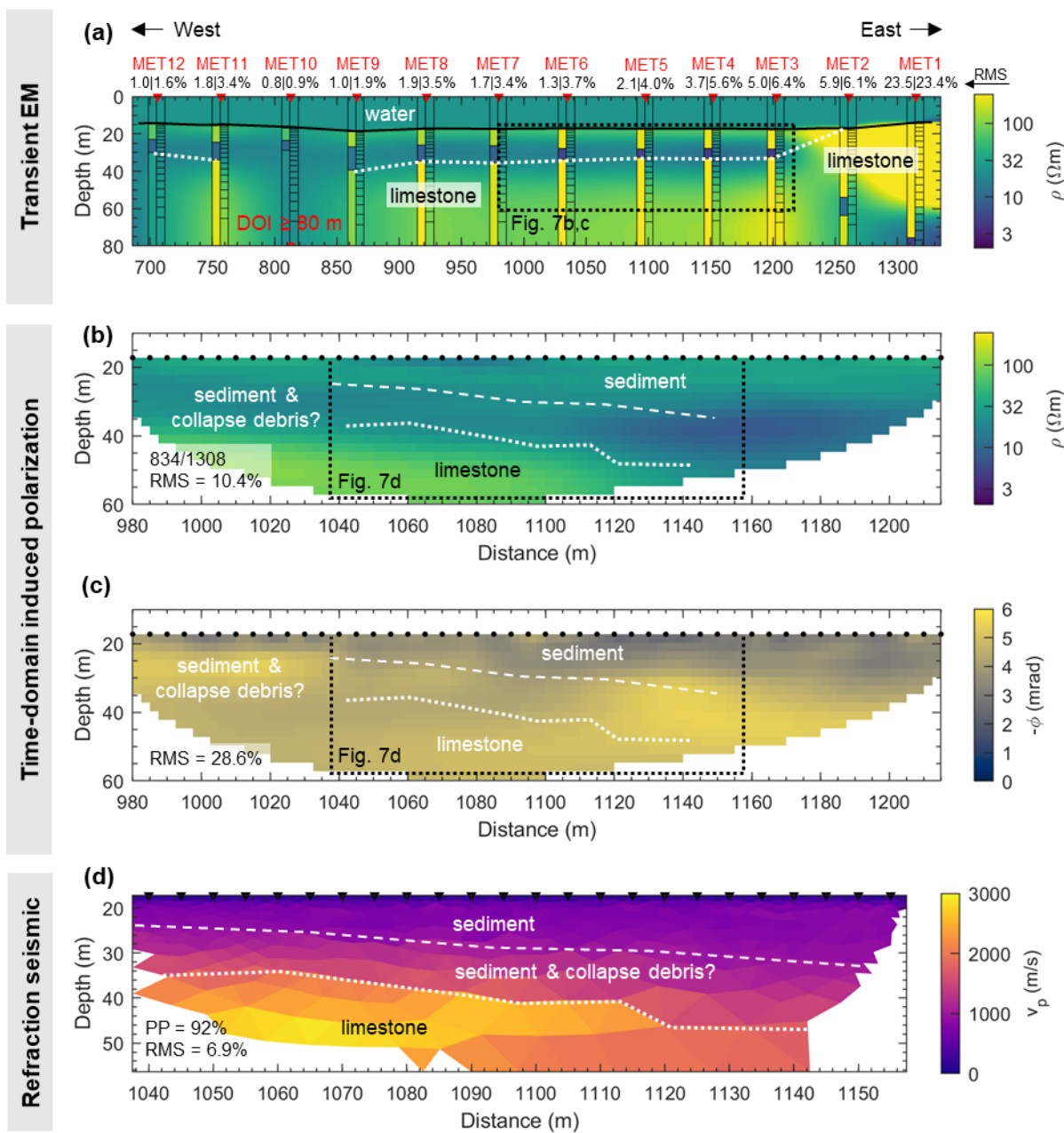

**Figure 7: Geophysical sections along Profile 4 of Lake Metzabok: (a) Interpolated TEM image based on smooth 1D models, left bar graphs show layered, right bar graphs smooth models; individual percentage RMS deviations are given for layered and smooth models, respectively. The black solid line in the section indicates the water-sediment contact, the dotted line the top of the limestone bedrock inferred from this image. (b) Electrical resistivity and (c) phase images including electrode positions (black dots along the surface), and the main lithological units as interpreted from the SRT image in (d) (dotted lines). Labels in the lower left corners of (b) and (c) represent the amount of data points used for the inversion compared to the total measured data (same for resistivity and phase) and the respective percentage RMS of the inversion. (d) Seismic refraction tomogram with main lithological units including picking percentage (PP) and RMS of the inversion.**



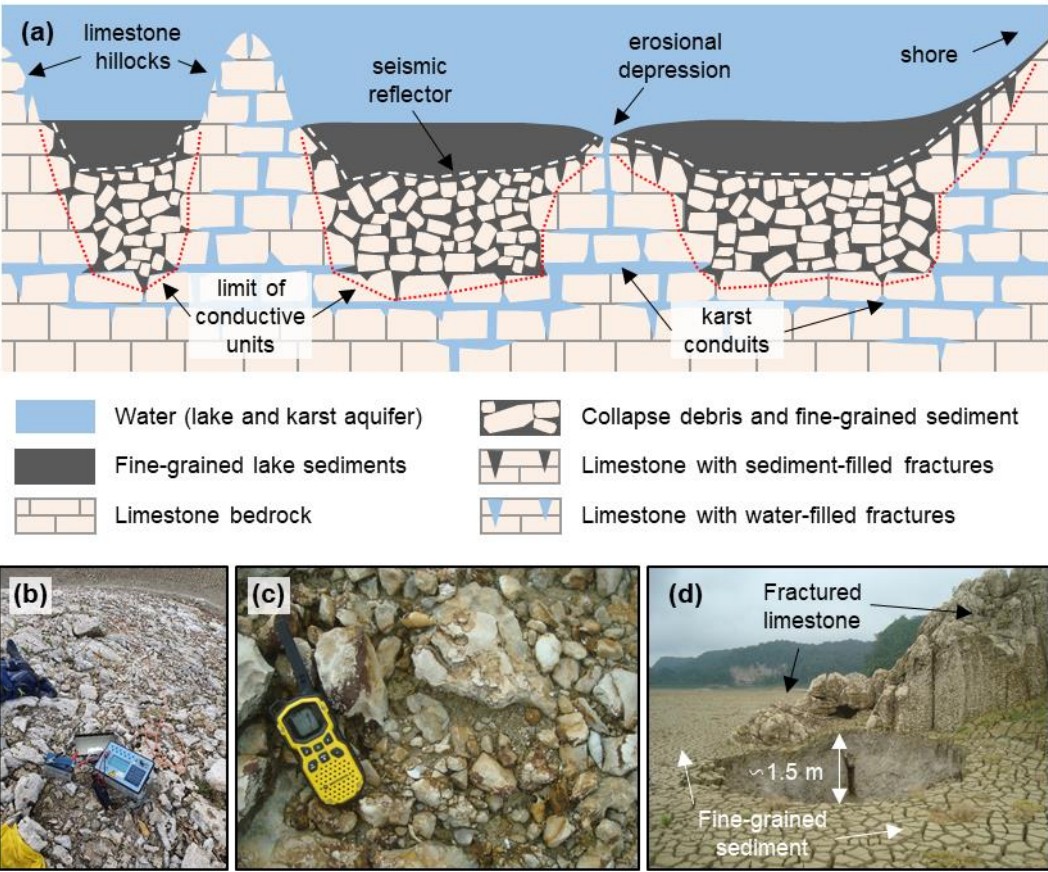

**Figure 8: (a) Schematic sketch summarizing the geological conditions below Lake Metzabok interpreted from the geophysical survey (details not drawn to scale). The limit between the fine-grained lake sediments and the collapse debris with sediment-filled interspaces indicated by the white dashed line stands out as a strong reflector in all sub-bottom profiler images. All units containing fine-grained sediment are characterized by low electrical resistivity values; a strong resistivity increase marks the upper limit of the limestone bedrock as indicated by the red dotted line. Photographs show (b) and (c) limestone debris with fine-grained sediment as well as (d) fractured limestone and fine lake sediments exposed during the low-level stands in October 2019.**


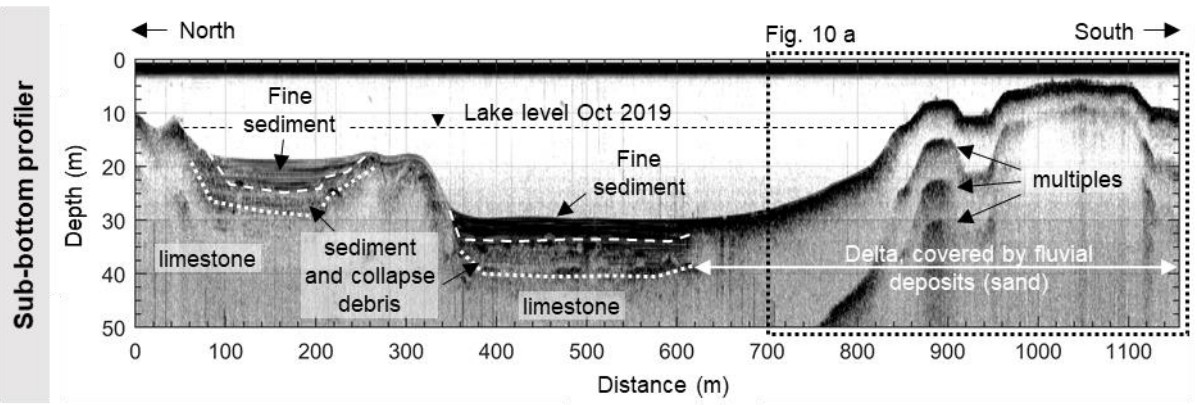

Figure 9: Long N-S oriented sub-bottom profiler section crossing the entire lake Tzibaná (Profile 5). The last approx. 450 m roughly coincide with the TEM section along Profile 6 in Fig. 10a (indicated by the dotted rectangle). The white dashed lines highlight the main reflector below the lake floor, which is interpreted as the lower limit of a fine-grained sediment layer. The white dotted lines enclose the zone of diffuse reflectivity associated with the collapsed, sediment-filled limestone. The black dashed line indicates the approximate lake level during the second field season in October 2020. The last 500 m of the profile correspond to the delta of the Nahá river, where a high reflectivity of the sand-covered lake floor results in the occurrence of strong multiples in the seismogram.

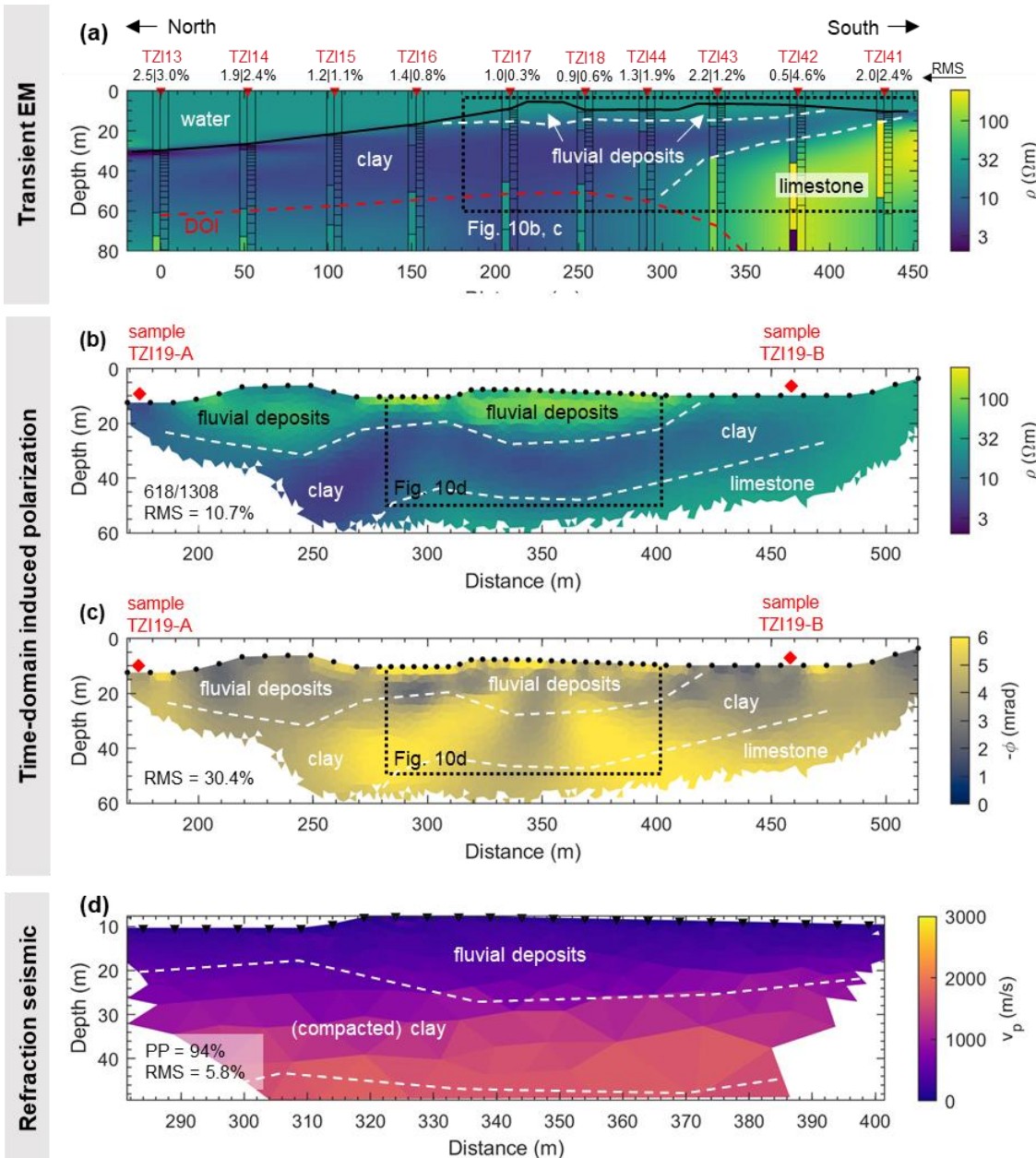

**Figure 10: Geophysical sections along Profile 6 of Lake Tzibaná: (a) Interpolated TEM image based on smooth 1D models, left bar graphs show layered, right bar graphs smooth models; individual percentage RMS deviations are given for layered and smooth models, respectively. The black solid line indicates the water-sediment contact, the dashed line the main lithological contacts inferred from this image. (b) Electrical resistivity and (c) phase images including electrode positions (black dots along the surface) and main lithological units. Red diamonds on the surface indicate the location of the sediments sampled for laboratory analyses. Labels in the lower left corners of (b) and (c) represent the amount of data points used for the inversion compared to the total measured data (same for resistivity and phase) and the respective percentage root mean square deviations (RMS) of the inversion. (d) Seismic refraction tomogram including picking percentage (PP) and RMS of the inversion, contacts of main lithological units (white dashed lines) taken from TDIP resistivity image.**

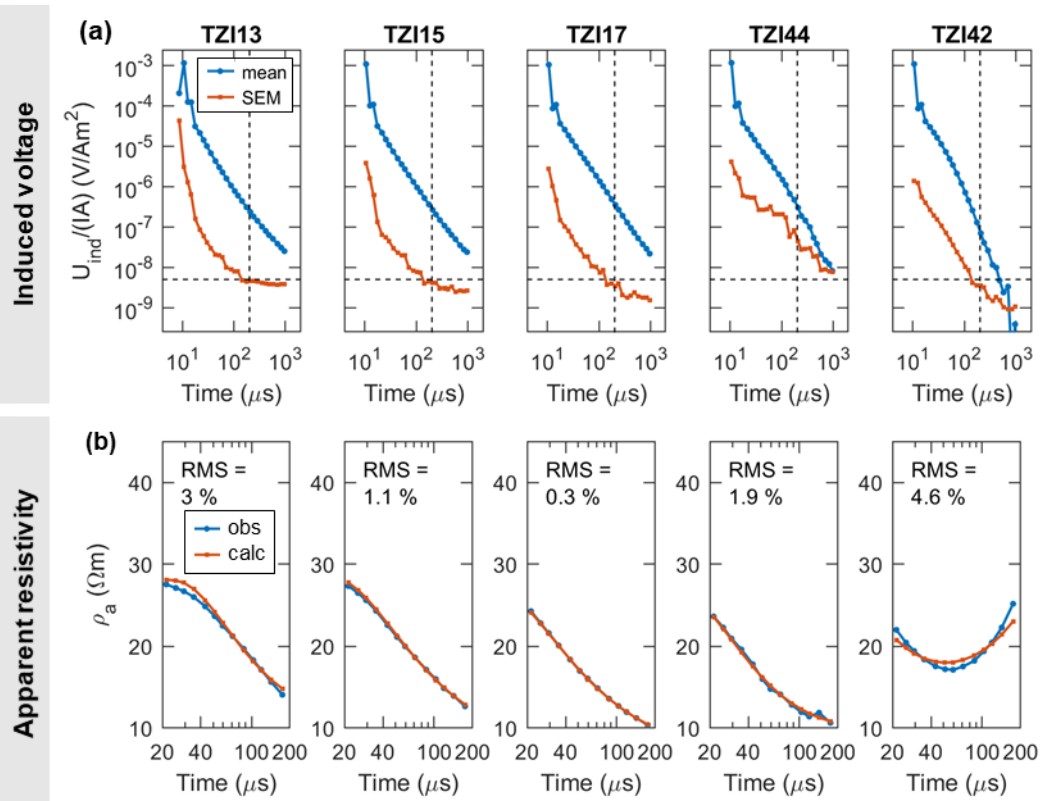

**Figure A1. a) Induced-voltage data of selected TEM soundings along Profile 6 of Lake Tzibaná. Induced voltages are normalized**
with the injected current and the loop area. The mean values of the stacked signal per time gate are shown in blue (circles) the
standard error of the mean in red (squares). Dashed lines indicate almost constant (exception: sounding TZI44) late-time error level
of $5 \cdot 10^{-9}$ V/Am² (horizontal lines) at 174.5 µs (vertical lines). b) Observed (blue circles) and calculated (red squares) apparent
resistivity curves of the same TEM soundings. The root-mean-square (RMS) errors of the individual model fits are indicated, too.
The corresponding inverted models (smooth models with 20 layers) are visualized in Fig. 9.

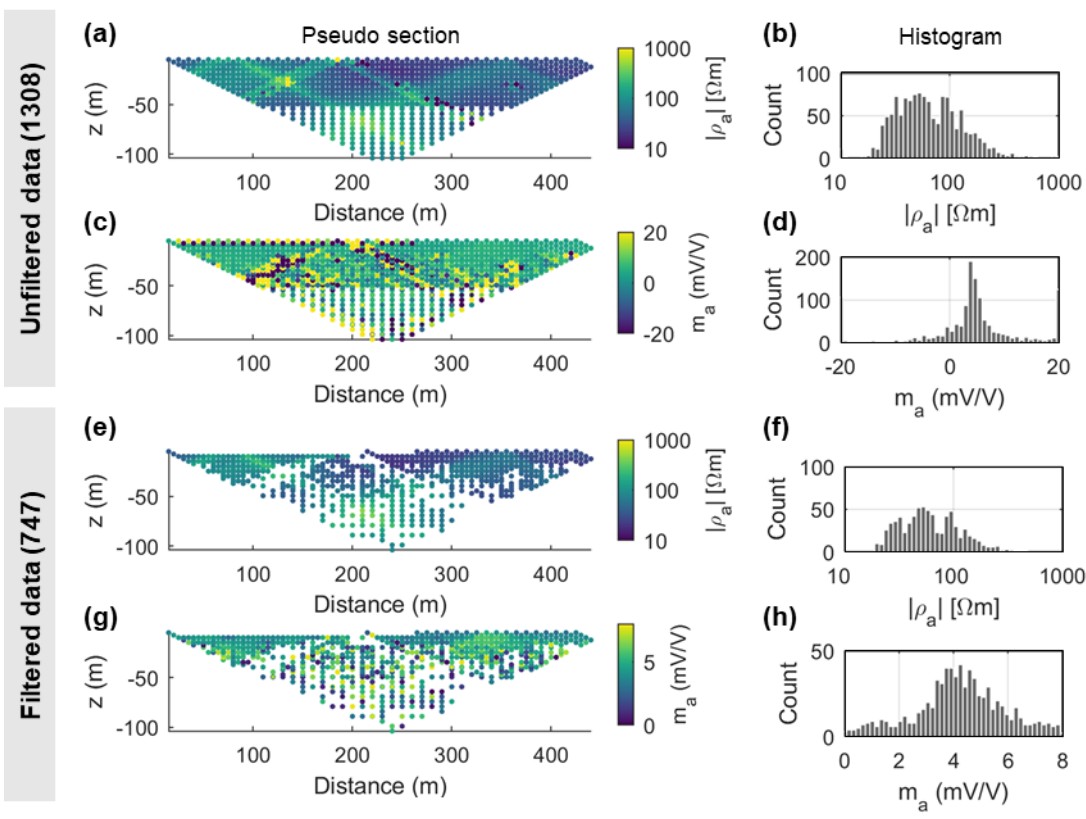

**Figure A2. Apparent resistivity and apparent chargeability pseudo sections (left column) and histograms (right column) of the TDIP measurement corresponding to the second part of roll-along Profile 1 of Lake Metzabok. The first two lines (a-d) show the unfiltered raw data set consisting of 1308 individual measurements, the last two lines (e-h) visualize the remaining 747 measurements after the application of the filters described in the main text.**

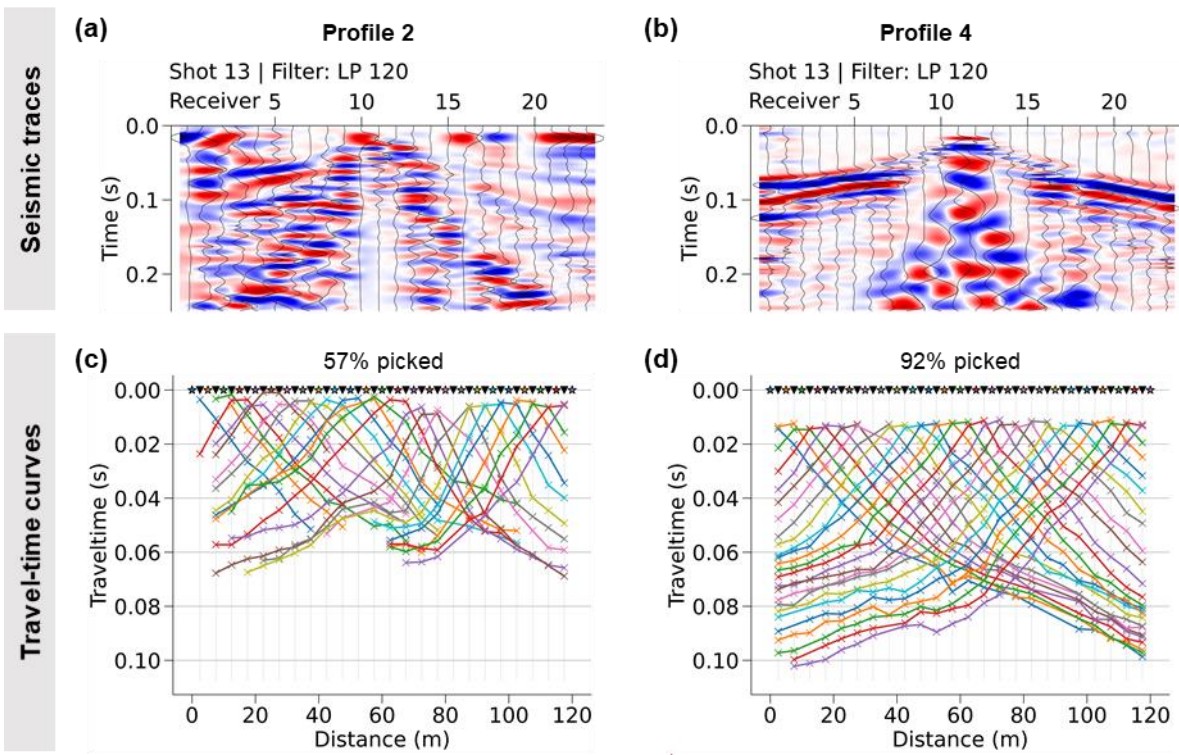

**Figure A3. Exemplary seismic traces (wiggle traces with variable density plots in the background) corresponding (a) to Profile 2 with a high noise level and (b) to Profile 4 with a much lower noise level. As a direct result of data quality, the travel-time curves (c) of Profile 2 (picking percentage 57%) are much less populated than those (d) of Profile 4 (92%). The loss of information particularly affects late travel times and thus significantly reduces the depth of investigation along noisy profiles.**
