# Peer review of "Integrated land and water-borne geophysical surveys shed light on the sudden drying of large karst lakes in southern Mexico"

_Solid Earth, 2020_

## Referee Comment (RC1) · Anonymous Referee #1 · 24 Jun 2020

**1   General comments**

The present manuscript is a solid compilation of results of geophysical measurements of two karst lakes in Mexico. The paper is well structured and the reader gets detailed insights into the extensive measurements with different geophysical methods. However, the paper strongly reminds of a project report. In my opinion a scientific paper in SE should go beyond a case study (maybe even with a more interesting title?). First, I would like to know more about the authors' motivation why this kind of lakes should be studied. In the introduction it is mentioned that it could be about choosing a suitable drilling location. This is not further discussed in the paper. I lack the approach how

these results can be transferred to other locations / problems. On the other hand, the fact that these karst lakes are falling dry is an extremely interesting point. Finally, this offers the geophysicists the possibility to repeat the measurements with a "covered" layer and to verify the results. This could be emphasized much more in the paper. The topic has much more potential to serve not only as a method comparison or case study.

**2 Specific comments**

- Chapter 3 deals in great detail with data acquisition and processing. All measurements of the applied wave and potential methods are clearly explained. In this as well as in the chapters 4 and 5 the insufficient signal-to-noise ratio is pointed out for some configurations. However, when evaluating the results, the reader is left in the dark when it comes to quantifying the error influences. This is an issue that urgently needs to be addressed - how much data could be included into the inversion process compared to the amount of measured data. How can the errors of the obtained models be estimated? (Example in line 182 - mean picking percents is ...)

- Is it possible that sample TSI19-A is influenced by higher limestone content? This would support the latter interpretation of field measurement results.

- In Line 214 ff. you mentioned underlain collapsed blocks - did you see some hints after the lakes are fallen dry?

- Line 229 and Fig. 4f: you do not interpret phase values for depth gt 50 m (due to insufficient data quality, ok - see first comment in list!), but than you should avoid to show this part of inversion model - it is more than the half of the picture!

- I am amazed by the variety of methods and the integrative approach for this survey. Only the complementary methods produce a comprehensive geological

model. I would not use chapter 5.3 as a confrontation of methods (title: seismic vs electrical methods) but rather promote these complementary techniques as a great advantage, the usage of the methods depends on the given situation and problem!

• Chapter conclusion should pick up some information from the introduction and give a broader (more general) summary at the end - how about the drilling, what is the take home message?

**3 Technical Corrections**

The pictures are generally of very good quality. Sometimes it is a bit confusing to recognize the correct position of the subprofiles (There are also different names for one and the same profile - to much information: example profile 1 aka L4NS aka MET19-1 MET19-2). Especially in Figure 5 it would help to use the same coordinates as in figure 4 (even if the profiles have an 10 m offset in EW direction). In Figure 4 the TDEM is slightly shifted in comparison to SBP.

---

## Referee Comment (RC2) · Pritam Yogeshwar (Referee) · 5 Jul 2020

**1. General Comments**
I have carefully and interestingly read and reviewed the manuscript (MS). The authors present an interesting multi-method case study on two lakes in Mexico using water borne geophysics. Especially the sudden lake water level drop as well as the application of multiple methods are highlights of this study. The results are very well prepared and technically on a high level.

Besides my positive impression, there are a few points to be addressed.

[Figure]

The authors present studies on two lakes - Metzabok and Tzibana. If feasible, I suggest to incorporate some more general discussion on how these lakes are connected. Are there any general conclusions/interpretations that apply to both lakes or is it even possible to connect the subsurface structure? For example, there is no clay interpreted in lake Metzabok whereas major parts of the subsurface in lake Tzibana is related to clay rich sediments.

It is not easy to find a red line in the MS. This is partly due to the fact that all profiles on lake Metzabok are discussed one by one. And, subsequently the results for lake Tzibana are shown. I suggest to strengthen the explicit motivation why both studies were performed. Possibly a road-map can be formulated indicating why which method was used and how the survey was designed to address the scientific questions. As I understand the study aims at few aspects (1) detect depth to bedrock (2) understand sudden lake level drop and related subsurface conditions such as Karst collapse and (3) combined interpretation of various methods (especially TD-IP phase data evaluation for the first time in sedimentary studies).

I suggest to elaborate more on the benefit of using TD-IP and evaluating the phase data for the two lake studies, since this is not very common. If feasible elaborate more in detail how the phase data relates to subsurface physical properties in general.

**2. Specific comments**

- I understand that the focus is on the geoscientific interpretation using a multidisciplinary approach. However, I do miss some technical aspects of the study with respect to method and inversion. For example some typical survey parameters (e.g. anchored or continuous TEM system; typical measurement errors).

  Moreover, there is currently no data visualized (Only the TD-IP lab data). I suggest to include a section with data, and possibly also with inversion model response (If feasible, for example in an appendix). Of course this should not distract from the study itself.

- From my experience, the TEMfast device sometimes shows significant distortions using small loop configurations. Did you observe any data distortions especially since a very small configurations was used? And, did you for example compare some land based soundings using a larger transmitter to validate that the very small layout gives correct transient data? In this respect, I also suggest to show at least some data.

- A conductor is indicated below the limestone towards the east in Fig. 7a. Please discuss this feature if it can be related to any geology such as fracture zones or if this is an artifact (probably related to distorted late time transient data). A slightly similar feature is also seen in Fig. 9 towards the south.

- Does the ZOND software actually invert for coincident loop or for a central loop receiver? For very early times the central loop transients differ from coincident loop data.

- The TEM data might be effected by 2D effects especially considering rather steep slope angles towards the edges. possibly include some discussion such as "multi-dimensional effects in TEM data were not considered as the TEM survey lines were not along strong bathymetry or steep slopes".

- P315 - Obviously the p-wave velocity is less than expected. Can you elaborate why a lower vp < 2000 m/s was observed in the SRT measurements.

- P350 - I suggest to include a table that summarizes the specifications of each method such as resolved physical parameter, DOI, pro/con of each method. Such a table would also summarize the used methods a bit and emphasize the integrative approach.

- P-365 - For TEM a water-depth of 20 m depending on the water conductivity is not necessarily a limitation. Please correct this statement.

- For all interpretation a smoothness constraint inversion is used. Do you expect a smooth transition from the sedimentary layers to the limestone. In this respect, is a smoothness constraint inversion appropriate to image the geological situation here?

- As water-borne TEM studies are still quite rare, I miss some references to recent water borne TEM studies. For example, we recently applied boat-towed TEM to image a hydrothermal target on the Azores. In this study we gathered around 600 soundings using the TEM system (initially developed by Mollidor et al.) in a continuous mode. There are also other very recent studies. These can be included as references, if the authors find them suitable:
  - Yogeshwar, P., Küpper, M., Tezkan, B., Rath, V., Kiyan, D., Byrdina, S., ... & Viveiros, F. (2020). Innovative boat-towed transient electromagnetics—Investigation of the Furnas volcanic lake hydrothermal system, Azores. Geophysics, 85(2), E41-E56.
  - Lane Jr, J. W., Briggs, M. A., Maurya, P. K., White, E. A., Pedersen, J. B., Auken, E., ... & Adams, R. (2020). Characterizing the diverse hydrogeology underlying rivers and estuaries using new floating transient electromagnetic methodology. Science of The Total Environment, 140074.

**3. Technical corrections**

The MS is very well written and the language is very good. All figures are well prepared with well readable fonts. Therefore, I only have a few technical corrections:

- P70 - the term reference data is misleading. I do not see that the data is actually used as reference data. Better - "additional/complementary data for comparison with the water borne data"

- Please check that all abbreviations are defined, e.g. ERT etc.

- P140 - explain or remove the skip parameters (skip-1 skip-2 etc.)
* * *

---

## Author Comment (AC1) · 27 Aug 2020

**Author response to Referee #1**

We would like to thank the referee for taking the time to review our manuscript and for providing thoughtful and substantial feedback, which we are sure will help us to substantially improve the manuscript.

In the following, reviewer comments are typeset in black, authors' answers (A) in blue, and planned modifications (M) in green.

**Anonymous Referee #1**

**1** General comments**

The present manuscript is a solid compilation of results of geophysical measurements of two karst lakes in Mexico. The paper is well structured and the reader gets detailed insights into the extensive measurements with different geophysical methods. However, the paper strongly reminds of a project report. In my opinion a scientific paper in SE should go beyond a case study (maybe even with a more interesting title?). First, I would like to know more about the authors' motivation why this kind of lakes should be studied. In the introduction it is mentioned that it could be about choosing a suitable drilling location. This is not further discussed in the paper. I lack the approach how these results can be transferred to other locations / problems. On the other hand, the fact that these karst lakes are falling dry is an extremely interesting point. Finally, this offers the geophysicists the possibility to repeat the measurements with a "covered" layer and to verify the results. This could be emphasized much more in the paper. The topic has much more potential to serve not only as a method comparison or case study.

**A: Motivation to study this kind of lakes**

We agree with the reviewer that besides the mere possibility to test a multi-methodological survey approach for the geophysical investigation of lakes, our motivation to study this kind of lakes remains unclear.

In general, the adaptation of geophysical techniques for the investigation of karst lakes is of relevance, given that ground and surface water in karst areas may react very sensible to even smaller variations of climatic conditions (e.g., precipitation, temperature, etc.). About 10% of the world's continental area is covered by karst and up to one quarter of the earth's population at least partly depends on drinking water from karst aquifer systems (e.g., Hartmann et al., 2014). Thus, karst water management will be important in the future and require reliable geological information. Where geological drillings are not available to obtain this information directly, geophysical methods have been used successfully to explore the subsurface of karst systems (Bechtel et al., 2007). Karst lakes are often in direct contact with the aquifer via karst conduits, which results in an increased vulnerability of both water bodies. This motivates the search for new, complementary geophysical exploration methods, which can be applied on land and on the water.

At the same time, paleoenvironmental studies, e.g. based on the analysis of lake sediments, can help to better understand the impact of a changing climate on karst systems. By extracting information on past variations from the sedimentary record of lakes, these studies shed light on the local links between climate and availability and quality of water of karst lakes and the connected karst aquifer. In order to obtain a continuous paleoenvironmental record, which goes as far back into the past as possible, the drilling sites for the extraction of lakes sediments have to be carefully selected. Detailed information on the sedimentary infill of the lakes, in particular on thickness, composition, and possible perturbations, helps to find a suitable location. Thus, a geophysical survey, which is able to provide such information, can greatly contribute to the success of the paleoenvironmental approach.

Field-observations and paleoenvironmental studies from the study area indicate that large seasonal lake-level variations are part of the nature of the lakes and even "catastrophic" events such as the one observed in 2019 might be recurrent (with a period >50 years). Thus, urgent questions in the study area are whether the sudden drainage of 2019 could be linked to recent climate change and whether it is possible that such events will occur more frequently in the future. Besides the ongoing paleoenvironmental investigation, a comprehensive geological picture of the lakes' geological substructure is essential to discuss possible draining mechanisms and their triggers.

Based on these considerations, the present geophysical study has three main objectives: (1) Determine suitable drilling locations to obtain complete, undisturbed, and far-reaching sedimentary sequences for paleoenvironmental studies, (2) provide basic knowledge on the geological substructure of the lakes (distribution and thickness of sediment cover, distribution of bare limestone rock with possible connectivity with the karst aquifer, etc.), and (3) adapt and test a multimethodological geophysical approach to achieve the first two objectives.

M: We will include this extended motivation into the revised version of the introduction of our manuscript and clearly state the three main objectives of the survey.

**A: Further discuss search for drilling location**

After having broadened the scope of the manuscript as sketched above, we will also discuss the selection of drilling sites as well as the geological information provided by the geophysical survey and its implications in more detail.

Based the results discussed so far, the thick (5-6 m according to the SBP image) and apparently undisturbed fine-grained lacustrine sediments along Profile 1 (between 450 and 550 m, as well as between 600 and 700 m) lend themselves for a successful paleolimnological perforation. A perforation at these sites would also have the potential to shed light on the real nature of the second layer and either verify or falsify our interpretation as debris-sediment mixture.

With a thickness of >50 m according to our electrical imaging results, the sedimentary cover along Profile 5 of Lake Tzibaná is thicker than the sediments of Lake Metzabok. However, our results also show that much of these sediments rather correspond to fluvial deposits of the river delta. In this depositional regime, we expect much higher rates of sedimentation, such that the mere sediment thickness does not necessarily imply an older record. In addition, a river delta is a much more complex system, in which sediments can be deposited, eroded, and redeposited repeatedly, which decreases the probability to obtain undistorted sediment records as encountered further offshore.

Or field observations and geophysical results also have implications for the general understanding of the geological setting of the two studied lakes. Obviously, large areas of Lake Metzabok are covered by a layer of fine-grained (clayey) sediments, which – in these areas – is sufficiently thick to act as a hydrological barrier between the lake and the surrounding and underlying karst. However, the remaining heavily fractured and uncovered limestone outcrops effectively connect the lake with the karst system. This conclusion is underscored by the velocity at which the two lakes drained practically simultaneously between February and July 2019. During this time, the water loss

amounted to 28,000 m3/day or 0.3 m3/s on average (Matti Altmann, personal communicatioin), which corresponds to the discharge of a large creek (e.g., McManamay and DeRolph, 2018).

While the interconnectivity between surface water and karst aquifer is well documented by field observations and further underpinned by our geophysical results, the actual mechanism as well as the cause of the sudden drainage remain unrevealed. The suddenness of the drainage (onset of discharge in the first days of February derived from satellite-image time series, Altmann, personal communication), indicates that one or more previously clogged karst conduits were unplugged around these dates. Planned time-series analyses of hydrological (e.g., lake water levels from satellite data) and meteorological data (precipitation, temperature, etc.) in combination with paleoenvironmental studies on sediment cores will possibly provide more hints on the mechanism and its triggers, and thus shed light on the question whether the drainage is influenced by climate variations.

M: We will include (and possibly further extend) this discussion in the revised version of our manuscript.

**A: Transfer of results to other locations and problems**

We consider the discussion of the combined application and interpretation of various geophysical methods on the two karst lakes itself a relevant reference for similar surveys at other locations. However, we agree with the reviewer in so far as the general importance of our findings can be developed in more detail.

M: Reviewer 2 suggested to provide a table that summarizes the used methods, as well as their advantages and limitations. Based on this table and the following key points we will extended our discussion of the general lessons learned for the geophysical investigation of lakes:

- TDIP on lakes: Based on the extended discussion of the IP response of lake sediments encourages by both reviewers, we will discuss further potential applications of this method in particular for shallow lakes with only a few meters of water column.
- TEM on lakes: In comparison to other floating TEM systems, the one used in this study is extremely light weight, cost-efficient, and quick to assemble. However, in terms of measurement noise and depth of investigation it provides comparable results as obtained using more sophisticated systems (e.g., Yogeshwar et al., 2020). We will compare the characteristics of our system in more detail with those of others in order to underpin this conclusion.
- Combination with seismics: The joint interpretation of electrical and seismic methods showed that combining "standard" seismic methods with electrical data greatly improves the scope of the geophysical investigation. While seismic methods are still the "standard" for the investigation of the sedimentary infill of lakes, our study shows that electrical methods perfectly complement the obtained information by providing hints to the actual composition and hydrological characteristics of seismic units.

**A: Emphasize more the fact that lakes fall dry and the positive implications for geophysics**

M: As suggested, in the revised version of the manuscript, we will put more emphasis on the positive implications of the drainage for our geophysical investigations. The occurrence of this event has really provided us with a unique opportunity to "repeat" measurements without a removed cover layer of up to 20 m water.

A: Not only method comparison or case study

We agree that study and topic have more potential, which will take advantage of by extending the scope of the manuscript as sketched above.

**2 Specific comments**

• Chapter 3 deals in great detail with data acquisition and processing. All measurements of the applied wave and potential methods are clearly explained. In this as well as in the chapters 4 and 5 the insufficient signal-to-noise ratio is pointed out for some configurations. However, when evaluating the results, the reader is left in the dark when it comes to quantifying the error influences. This is an issue that urgently needs to be addressed - how much data could be included into the inversion process compared to the amount of measured data. How can the errors of the obtained models be estimated? (Example in line 182 - mean picking percents is ...)

M: We will describe the determination of the measurement error for the TEM, TDIP, and SRT data and include exemplary data and error visualizations in a new appendix. Data quality and its general implications for the depth of investigation will also be discussed in the main text. The model misfit of the final inverted models will be included into the figures showing the imaging results.

TEM. The determination and assessment of the measurement error for the TEM-soundings is discussed in more detail in the answers to the comments of Reviewer 2.

TDIP. As described in the main text of the manuscript, the TDIP data is filtered based on the apparent resistivity and phase data. In a first step, erroneous measurements with apparent resistivity values  $\leq 0 \Omega m$  and/or apparent phase values  $\leq 0 mrad$  are removed. Based on the visual assessment of the raw data pseudo sections (see Figure R1.1 for an example), in a second filtering step, measurements with apparent phase values > 8 mrad are removed as further outliers. The selection of the upper limit of 8 mrad for the apparent phase values is based on the observation of a narrow distribution of "physically meaningful" phase values in the apparent-phase histograms (see Figure R1.1). In the case of the second part of the roll-along profile 1 of Lake Metzabok, this filtering results in a reduction of the TDIP data set to 57% of the original data set. This loss of data is related to a relatively poor data quality of the phase measurements along this long line (470 m length, 10 m electrode spacing) and is expected to significantly reduce the depth of investigation of the phase images. Data sets of shorter profiles with a reduced electrode spacing (5 m in the case of profiles 2-5) are less affected by phase noise, which results in a higher percentage of useful data (up to ~88% in the case of Profile 3 of Lake Metzabok.

In the revised version of our manuscript, we will include an exemplary TDIP data set (in the new appendix) and provide information on the percentage of useful data as well as the individual model misfits (in terms of the RMS error) of all resistivity and phase images. In the main text, we will briefly discuss the effect of individual data quality on the depth of investigation of each image.

**Figure R1.1.** Apparent resistivity and apparent phase pseudo sections (left column) and histograms (right column). The first two lines show unfiltered raw data set consisting of 1308 measurements, the last two lines show the remaining data after the application of the filters described in the main text (747 measurements).

SRT. In order to provide more detail on the data quality of the SRT measurements, in the new appendix, we will provide exemplary visualizations of first-arrival picks for one relatively clean and one relatively noisy measurement. Collected with 24 geophones and 25 shot positions, each tomographic data set consists of a total of 600 seismic traces. The picking percentage reflects the number of traces, for which a first arrival can be identified, and serves as a measure for data quality. Figure R1.2a shows data and travel time curves for Profile 2 and Profile 4 of Lake Metzabok. A low data quality of Profile 2 data results in a low picking percentage (341 out of a total of 600 traces) and mainly affects long-offset data reducing the expected depth of investigation. In comparison, the SRT data collected along Profile 4 is much cleaner (552 out of 600 traces), which results in a larger depth of exploration.

In the revised version of out manuscript, we will include exemplary SRT data and travel-time curves (in the new appendix) and provide information on picking percentages for all SRT profiles. We will also provide information on the individual model misfits (in terms of the RMS error) of the inverted velocity images. In the main text, we will link the depth of investigation of the individual images to the data quality (i.e., the picking percentage).

---

## Author Comment (AC2) · 27 Aug 2020

**Author response to Referee #2 (Pritam Yogeshwar, University of Cologne)**

Thank you, Pritam, for taking your time to review our manuscript and for your detailed and constructive feedback – especially on the TEM method – which we will surely help us to improve the manuscript a lot!

In the following, reviewer comments are typeset in black, authors' answers (A) in blue, and planned modifications (M) in green.

**Pritam Yogeshwar (Referee)**

yogeshwar@geo.uni-koeln.de

**1.** General Comments**

I have carefully and interestingly read and reviewed the manuscript (MS). The authors present an interesting multi-method case study on two lakes in Mexico using water borne geophysics. Especially the sudden lake water level drop as well as the application of multiple methods are highlights of this study. The results are very well prepared and technically on a high level. Besides my positive impression, there are a few points to be addressed.

The authors present studies on two lakes - Metzabok and Tzibana. If feasible, I suggest to incorporate some more general discussion on how these lakes are connected. Are there any general conclusions/interpretations that apply to both lakes or is it even possible to connect the subsurface structure? For example, there is no clay interpreted in lake Metzabok whereas major parts of the subsurface in lake Tzibana is related to clay rich sediments.

A: We agree. Resulting from the very limited information provided on Lake Tzibaná, it does not become clear, how the two neighboring lakes are connected. Figure R2.1 shows the result of one long SBP transect crossing Lake Tzibaná and the delta of the Nahá river from North to South. From this line, it becomes clear that outside the delta with it sandy sediments, the deeper part of Lake Tzibaná shows a similar geology as Lake Metzabok (flat sediment-covered lake bottom). However, the flat parts of Lake Tzibaná, which are expected to be more comparable with Metzabok, are much deeper than Metzabok and were also not accessible (still water covered) during the second field season, such that no additional data could be collected directly on the lake bottom.

*Figure R2.1.* Long SBP profile crossing Lake Tzibaná from North to South. The last ~400 m roughly coincide with the position of Profile 5 (indicated by dotted rectangle). Below the sandy delta

**sediments, no further reflectors can be observed. As in the case of Lake Metzabok, also the flat zones of Lake Tzibaná are covered by an at least 5 m thick layer of (fine-grained) lake sediment.**

M: We will include this SBP profile in the revised version of the manuscript and provide a more complete discussion of the common features and differences between the two studied lakes.

It is not easy to find a red line in the MS. This is partly due to the fact that all profiles on lake Metzabok are discussed one by one. And, subsequently the results for lake Tzibana are shown. I suggest to strengthen the explicit motivation why both studies were performed. Possibly a road-map can be formulated indicating why which method was used and how the survey was designed to address the scientific questions. As I understand the study aims at few aspects (1) detect depth to bedrock (2) understand sudden lake level drop and related subsurface conditions such as Karst collapse and (3) combined interpretation of various methods (especially TD-IP phase data evaluation for the first time in sedimentary studies).

A: We agree. The focus of the manuscript in its current form is the comparison of methods, which is developed and discussed step by step along the profiles. Both received reviews demand for a broadening of the scope by improving the description of the more general scientific motivation and the inclusion of a more rigorous geological/hydrogeological contextualization. In our response to reviewer 1, we develop concrete idea of how to respond to this suggestion, which also includes a "road-map", which is similar to the one suggested here.

I suggest to elaborate more on the benefit of using TD-IP and evaluating the phase data for the two lake studies, since this is not very common. If feasible elaborate more in detail how the phase data relates to subsurface physical properties in general.

A: We also see the benefit of discussing the IP response of lake sediments in more detail.

M: Motivated by the question of Reviewer 1, whether the sample TSI19-A might contain a larger fraction of limestone, we will extend our discussion of the IP response of lake sediments in the results and interpretation section and possibly include selected geochemical laboratory data for this purpose. Based on this discussion, we will also be able to draw some (preliminary) conclusions regarding the possible contribution of IP data for the study of lakes and lake sediments.

**2. Specific comments**

• I understand that the focus is on the geoscientific interpretation using a multidisciplinary approach. However, I do miss some technical aspects of the study with respect to method and inversion. For example, some typical survey parameters (e.g. anchored or continuous TEM system; typical measurement errors).

A: We recognize that we might have overshot the target of providing very brief descriptions of the various methods used in this multi-method survey. In the following, we provide some additional information on the TEM measurement setup and the inversion approach.

During the survey, the TEM system was towed from station to station. The electromotor of the rubber boat was only used for navigation between sounding locations and remained turned off during the measurements themselves. Because we did not use any anchors, depending on the wind conditions at the sounding sites, the loop slowly drifted during the measurements resulting in maximum estimated displacements of 1 - 2 times the loop diameter (i.e., ~40 m).

For the transient length of 1024  $\mu$ s used for our measurements, the TEM-FAST48 records 64 transients, which are analogously averaged by the hardware. For one sounding measurement, this

basic measuring cycle is repeated  $n \times 13$  times. For n = 4 (our measurements), this results in 52 repetitions of the basic cycle (and a total of 3328 effective stacks), which are used to compute the impulse response by digital averaging and to determine the measurement error as the standard error of the mean (SEM). As the exemplary data in Figure R2.2 shows, for the latest time gates used for the inversion (around 200 µs), the SEM is  $\leq 2 \cdot 10^{-6}$  V/A (or  $\leq 5 \cdot 10^{-9}$  V/Am2).

---

## Author Response (AR1)

**Author response to Referee #1**

**Update submitted along with the revised manuscript**

We would like to thank the referee for taking the time to review our manuscript and for providing thoughtful and substantial feedback, which we are sure will help us to substantially improve the manuscript.

In the following, reviewer comments are typeset in black, authors' answers (A) in blue, and realized modifications (M) in green.

Anonymous Referee #1

**1 General comments**

The present manuscript is a solid compilation of results of geophysical measurements of two karst lakes in Mexico. The paper is well structured and the reader gets detailed insights into the extensive measurements with different geophysical methods. However, the paper strongly reminds of a project report. In my opinion a scientific paper in SE should go beyond a case study (maybe even with a more interesting title?). First, I would like to know more about the authors' motivation why this kind of lakes should be studied. In the introduction it is mentioned that it could be about choosing a suitable drilling location. This is not further discussed in the paper. I lack the approach how these results can be transferred to other locations / problems. On the other hand, the fact that these karst lakes are falling dry is an extremely interesting point. Finally, this offers the geophysicists the possibility to repeat the measurements with a "covered" layer and to verify the results. This could be emphasized much more in the paper. The topic has much more potential to serve not only as a method comparison or case study.

A: Motivation to study this kind of lakes

M: New title: "Integrated land and water-borne geophysical surveys shed light on the sudden drying of large karst lakes in southern Mexico"

A: Motivation to study this kind of lakes

We agree with the reviewer that besides the mere possibility to test a multi-methodological survey approach for the geophysical investigation of lakes, our motivation to study this kind of lakes remains unclear.

In general, the adaptation of geophysical techniques for the investigation of karst lakes is of relevance, given that ground and surface water in karst areas may react very sensible to even smaller variations of climatic conditions (e.g., precipitation, temperature, etc.). About 10% of the world's continental area is covered by karst and up to one quarter of the earth's population at least partly depends on drinking water from karst aquifer systems (e.g., Hartmann et al., 2014). Thus, karst water management will be important in the future and require reliable geological information. Where geological drillings are not available to obtain this information directly, geophysical methods have been used successfully to explore the subsurface of karst systems (Bechtel et al., 2007). Karst lakes are often in direct contact with the aquifer via karst conduits, which results in an increased vulnerability of both water bodies. This motivates the search for new, complementary geophysical exploration methods, which can be applied on land and on the water.

At the same time, paleoenvironmental studies, e.g. based on the analysis of lake sediments, can help to better understand the impact of a changing climate on karst systems. By extracting information on past variations from the sedimentary record of lakes, these studies shed light on the local links between climate and availability and quality of water of karst lakes and the connected karst aquifer. In order to obtain a continuous paleoenvironmental record, which goes as far back into the past as possible, the drilling sites for the extraction of lakes sediments have to be carefully selected. Detailed information on the sedimentary infill of the lakes, in particular on thickness, composition, and possible perturbations, helps to find a suitable location. Thus, a geophysical survey, which is able to provide such information, can greatly contribute to the success of the paleoenvironmental approach.

Field-observations and paleoenvironmental studies from the study area indicate that large seasonal lake-level variations are part of the nature of the lakes and even "catastrophic" events such as the one observed in 2019 might be recurrent (with a period >50 years). Thus, urgent questions in the study area are whether the sudden drainage of 2019 could be linked to recent climate change and whether it is possible that such events will occur more frequently in the future. Besides the ongoing paleoenvironmental investigation, a comprehensive geological picture of the lakes´ geological substructure is essential to discuss possible draining mechanisms and their triggers.

Based on these considerations, the present geophysical study has three main objectives: (1) Determine suitable drilling locations to obtain complete, undisturbed, and far-reaching sedimentary sequences for paleoenvironmental studies, (2) provide basic knowledge on the geological substructure of the lakes (distribution and thickness of sediment cover, distribution of bare limestone rock with possible connectivity with the karst aquifer, etc.), and (3) adapt and test a multi-methodological geophysical approach to achieve the first two objectives.

M: We included this extended motivation into the revised version of the introduction of our manuscript, where we also clearly state the three main objectives of the survey. Corresponding adaptations were made in the abstract, too.

Along with the corresponding changes, we removed the references to Cohuo et al. (2018), Pérez et al. (2011), and Sigala et al. (2017) and included new references to Vázquez-Molina et al. (2016), Pérez et al, (2020), Medina-Elizalde and Rohling (2012), Echeverría Galindo et al. (2019), and Charqueño Celis et al. (2019).

A: Further discuss search for drilling location

After having broadened the scope of the manuscript as sketched above, we will also discuss the selection of drilling sites as well as the geological information provided by the geophysical survey and its implications in more detail.

Based the results discussed so far, the thick (5-6 m according to the SBP image) and apparently undisturbed fine-grained lacustrine sediments along Profile 1 (between 450 and 550 m, as well as between 600 and 700 m) lend themselves for a successful paleolimnological perforation. A perforation at these sites would also have the potential to shed light on the real nature of the second layer and either verify or falsify our interpretation as debris-sediment mixture. With a thickness of >50 m according to our electrical imaging results, the sedimentary cover along Profile 5 of Lake Tzibaná is thicker than the sediments of Lake Metzabok. However, our results also show that much of these sediments rather correspond to fluvial deposits of the river delta. In this depositional regime, we expect much higher rates of sedimentation, such that the mere sediment thickness does not necessarily imply an older record. In addition, a river delta is a much more complex system, in

which sediments can be deposited, eroded, and redeposited repeatedly, which decreases the probability to obtain undistorted sediment records as encountered further offshore.

Or field observations and geophysical results also have implications for the general understanding of the geological setting of the two studied lakes. Obviously, large areas of Lake Metzabok are covered by a layer of fine-grained (clayey) sediments, which – in these areas – is sufficiently thick to act as a hydrological barrier between the lake and the surrounding and underlying karst. However, the remaining heavily fractured and uncovered limestone outcrops effectively connect the lake with the karst system. This conclusion is underscored by the velocity at which the two lakes drained practically simultaneously between February and July 2019. During this time, the water loss amounted to 28,000 m³/day or 0.3 m³/s on average (Matti Altmann, personal communicatioin), which corresponds to the discharge of a large creek (e.g., McManamay and DeRolph, 2018).

While the interconnectivity between surface water and karst aquifer is well documented by field observations and further underpinned by our geophysical results, the actual mechanism as well as the cause of the sudden drainage remain unrevealed. The suddenness of the drainage (onset of discharge in the first days of February derived from satellite-image time series, Altmann, personal communication), indicates that one or more previously clogged karst conduits were unplugged around these dates. Planned time-series analyses of hydrological (e.g., lake water levels from satellite data) and meteorological data (precipitation, temperature, etc.) in combination with paleoenvironmental studies on sediment cores will possibly provide more hints on the mechanism and its triggers, and thus shed light on the question whether the drainage is influenced by climate variations.

M: We included and further extended this discussion as a new section "5.1 Identification of suitable drilling locations" into the revised version of our manuscript.

A: Transfer of results to other locations and problems

We consider the discussion of the combined application and interpretation of various geophysical methods on the two karst lakes itself a relevant reference for similar surveys at other locations. However, we agree with the reviewer in so far as the general importance of our findings can be developed in more detail.

M: Reviewer 2 suggested to provide a table that summarizes the used methods, as well as their advantages and limitations. Based on this table, we rewrote large parts of the discussion section under the new subsection title "5.3 Lessons learned from implementing a multi-methodological approach for lake-bottom reconnaissance".

A: Emphasize more the fact that lakes fall dry and the positive implications for geophysics

M: As suggested, in the revised version of the manuscript, we put more emphasis on the positive implications of the drainage for our geophysical investigations. The occurrence of this event has really provided us with a unique opportunity to "repeat" measurements without a removed cover layer of up to 20 m water.

A: Not only method comparison or case study

We agree that study and topic have more potential.

M: We took advantage of this potential by extending the scope of the manuscript as sketched above.

**2 Specific comments**

• Chapter 3 deals in great detail with data acquisition and processing. All measurements of the applied wave and potential methods are clearly explained. In this as well as in the chapters 4 and 5 the insufficient signal-to-noise ratio is pointed out for some configurations. However, when evaluating the results, the reader is left in the dark when it comes to quantifying the error influences. This is an issue that urgently needs to be addressed - how much data could be included into the inversion process compared to the amount of measured data. How can the errors of the obtained models be estimated? (Example in line 182 - mean picking percents is ...)

M: We now describe the error and processing of TEM, TDIP, and SRT data in more detail and included exemplary data and error visualizations in a new appendix. Data quality and its general implications for the depth of investigation are also discussed in more detail. The model misfit of the final inverted models was included into all figures showing imaging results.

• Is it possible that sample TSI19-A is influenced by higher limestone content? This would support the latter interpretation of field measurement results.

A: A more detailed laboratory analysis of the sediment samples is still underway. Thus, we have not yet been able to understand the obvious deviation of the TDIP response of sample TSI19-A.

Preliminary results do not indicate a different mineralogical composition of sample TSI19-A compared to the rest of the samples. X-ray powder diffraction analyses show similar concentrations of dolomite (calcium magnesium carbonate) and calcite (calcium carbonate) in all six samples. However, the geochemical analysis does show a significantly increased total organic carbon (TOC) and carbon-to-nitrogen ratio (C/N) of the sample TSI19-A compared to the other five samples. Together, the high levels of these two parameters point to a larger fraction of organic matter from terrestrial sources (i.e., land-based plants), while the smaller amount of organic matter of the other five samples probably stems from algal plants.

Both samples TSI19-A and TSI19-B were collected on the exposed delta of the Nahá river. However, the location TSI19-A seems to have received more fluvial deposits due to its position closer to the estuary of the river, while location TSI19-B corresponds to a residual body water (i.e., in a predominantly aquatic environment) situated within the delta. In fact, in Figure 9, we can see that TSI19-A is located on the northernmost extension of the unit labelled as fluvial deposits, while TSI19-B is located in a (partly water-filled) depression underlain by the fine-grained (clayey) lake sediments. Thus the deviation of the TDIP response of sample TSI19-A is rather linked to the source of the sediment (fluvial vs. lacustrine) than to its mineralogical composition (e.g., limestone content) as might be assumed based in the TDIP field survey, where particularly higher phase responses can be related to the limestone bedrock.

M: We included a summary of this discussion into the revised manuscript in order to make clear that the higher phase and resistivity values of the sample TSI19-A are rather related to the depositional regime than the limestone content.

• In Line 214 ff. you mentioned underlain collapsed blocks - did you see some hints after the lakes are fallen dry?

A: The (few) uncovered limestone outcrops show both highly fractured limestone as well as limestone debris. However, the presence of limestone debris below the (thick) sediment cover has not been validated independently in the field. Thus, the underlying collapsed bedrock is clearly an interpretation, which is solely based on (i) a reasonable conceptual model of the evolution of the

lakes from collapsed bedrock and (ii) the layered resistivity structure below the flat sediment-filled parts of lake Metzabok (pure sediments, sediment-debris mixture, bedrock).

M: We included an additional Figure (Fig. 8b-d, revised manuscript) showing superficial expressions of the interpreted geological units, in particular the limestone debris and the fractured limestone.

• Line 229 and Fig. 4f: you do not interpret phase values for depth gt 50 m (due to insufficient data quality, ok - see first comment in list!), but than you should avoid to show this part of inversion model - it is more than the half of the picture!

M: We blanked less sensitive areas (cumulative normalized sensitivity smaller than 1/100 of the largest sensitivity) of all TDIP imaging results and adjusted the maximum depths of the corresponding figures accordingly.

• I am amazed by the variety of methods and the integrative approach for this survey. Only the complementary methods produce a comprehensive geological model. I would not use chapter 5.3 as a confrontation of methods (title: seismic vs electrical methods) but rather promote these complementary techniques as a great advantage, the usage of the methods depends on the given situation and problem!

M: We removed section "5.3 Seismic vs. electrical methods" and reorganized the entire part of the discussion section dealing with the evaluation of the individual methods. We also highlight the advantage of using complementary methods and data to obtain a comprehensive geological model.

• Chapter conclusion should pick up some information from the introduction and give a broader (more general) summary at the end - how about the drilling, what is the take home message?

M: As suggested, the revised conclusion section became broader and more general summary. In particular, we will include conclusive statements regarding the findings with respect to a more general geological interpretation (also suggested by the second reviewer), the selection of drilling locations for paleoenvironmental studies, and the lessons learned from the evaluation of the individual methods.

**3 Technical Corrections**

The pictures are generally of very good quality. Sometimes it is a bit confusing to recognize the correct position of the subprofiles (There are also different names for one and the same profile - to much information: example profile 1 aka L4NS aka MET19-1 MET19-2). Especially in Figure 5 it would help to use the same coordinates as in figure 4 (even if the profiles have an 10 m offset in EW direction). In Figure 4 the TDEM is slightly shifted in comparison to SBP.

M: We removed the labels referring to the IDs of the data set published along with the manuscript (e.g., L4NS, MET19-1, etc.). In the revised version of the manuscript, in Figure 5, we use the same coordinates as in Figure 4. The TDIP images in Figures 4f and 4g have an offset of about 25 m with respect to the SBP profile in Figure 4e. We included dotted arrows to indicate the position of the TDIP profile with respect to the SBP image.

**Additional references for this author response (all other references included in the manuscript)**

McManamay, R. A., & DeRolph, C. R. (2019). A stream classification system for the conterminous United States. Scientific data, 6, 190017.

**Author response to Referee #2 (Pritam Yogeshwar, University of Cologne)**

**Update submitted along with the revised manuscript**

Thank you, Pritam, for taking your time to review our manuscript and for your detailed and constructive feedback – especially on the TEM method – which we will surely help us to improve the manuscript a lot!

In the following, reviewer comments are typeset in black, authors' answers (A) in blue, and planned modifications (M) in green.

Pritam Yogeshwar (Referee)

yogeshwar@geo.uni-koeln.de

**1. General Comments**

I have carefully and interestingly read and reviewed the manuscript (MS). The authors present an interesting multi-method case study on two lakes in Mexico using water borne geophysics. Especially the sudden lake water level drop as well as the application of multiple methods are highlights of this study. The results are very well prepared and technically on a high level. Besides my positive impression, there are a few points to be addressed.

The authors present studies on two lakes - Metzabok and Tzibana. If feasible, I suggest to incorporate some more general discussion on how these lakes are connected. Are there any general conclusions/interpretations that apply to both lakes or is it even possible to connect the subsurface structure? For example, there is no clay interpreted in lake Metzabok whereas major parts of the subsurface in lake Tzibana is related to clay rich sediments.

A: We agree. Resulting from the very limited information provided on Lake Tzibaná, it does not become clear, how the two neighboring lakes are connected. The new Fig. 9 of the revised manuscript shows the result of one long SBP transect crossing Lake Tzibaná and the delta of the Nahá river from North to South. From this line, it becomes clear that outside the delta with it sandy sediments, the deeper part of Lake Tzibaná shows a similar geology as Lake Metzabok (flat sediment-covered lake bottom). However, the flat parts of Lake Tzibaná, which are expected to be more comparable with Metzabok, are much deeper than Metzabok and were also not accessible (still water covered) during the second field season, such that no additional data could be collected directly on the lake bottom.

M: We included a new SBP profile in the revised version of the manuscript and provide a more complete discussion of the common features and differences between the two studied lakes.

It is not easy to find a red line in the MS. This is partly due to the fact that all profiles on lake Metzabok are discussed one by one. And, subsequently the results for lake Tzibana are shown. I suggest to strengthen the explicit motivation why both studies were performed. Possibly a road-map can be formulated indicating why which method was used and how the survey was designed to address the scientific questions. As I understand the study aims at few aspects (1) detect depth to bedrock (2) understand sudden lake level drop and related subsurface conditions such as Karst collapse and (3) combined interpretation of various methods (especially TD-IP phase data evaluation for the first time in sedimentary studies).

A: We agree. The focus of the manuscript in its current form is the comparison of methods, which is developed and discussed step by step along the profiles. Both received reviews demand for a broadening of the scope by improving the description of the more general scientific motivation and the inclusion of a more rigorous geological/hydrogeological contextualization.

M: In response to these suggestions, we reworked the abstract, the introduction section, the discussion section, and the conclusions. We explicitly include a "road-map" similar to the one suggested here.

I suggest to elaborate more on the benefit of using TD-IP and evaluating the phase data for the two lake studies, since this is not very common. If feasible elaborate more in detail how the phase data relates to subsurface physical properties in general.

A: We also see the benefit of discussing the IP response of lake sediments in more detail.

M: Mainly guided by the question of Reviewer 1, whether the sample TSI19-A might contain a larger fraction of limestone, we extended our discussion of the IP response of lake sediments in the results and interpretation section and the discussion section.

**2. Specific comments**

• I understand that the focus is on the geoscientific interpretation using a multidisciplinary approach. However, I do miss some technical aspects of the study with respect to method and inversion. For example, some typical survey parameters (e.g. anchored or continuous TEM system; typical measurement errors).

A: We recognize that we might have overshot the target of providing very brief descriptions of the various methods used in this multi-method survey. In the following, we provide some additional information on the TEM measurement setup and the inversion approach.

During the survey, the TEM system was towed from station to station. The electromotor of the rubber boat was only used for navigation between sounding locations and remained turned off during the measurements themselves. Because we did not use any anchors, depending on the wind conditions at the sounding sites, the loop slowly drifted during the measurements resulting in maximum estimated displacements of $1 - 2$ times the loop diameter (i.e., ~40 m).

For the transient length of 1024 μs used for our measurements, the TEM-FAST48 records 64 transients, which are analogously averaged by the hardware. For one sounding measurement, this basic measuring cycle is repeated $n \times 13$ times. For $n = 4$ (our measurements), this results in 52 repetitions of the basic cycle (and a total of 3328 effective stacks), which are used to compute the impulse response by digital averaging and to determine the measurement error as the standard error of the mean (SEM). As the exemplary data the new Figure A1 shows, for the latest time gates used for the inversion (around 200 μs), the SEM is $\lesssim 2 \cdot 10^{-6}$ V/A (or $\lesssim 5 \cdot 10^{-9}$ V/Am²).

We use the conventional 1D smoothness-constraint inversion approach implemented in the software ZondTEM1d (Kaminsky, personal communication) to interpret the TEM soundings. The software supports arbitrary shaped loops, the vertices of which can be defined independently for transmitter and receiver. This warrants a correct interpretation of our coincident loop transient data. While our measurements were carried out with a circular loop (transmitter and receiver) of 22.9 m diameter (area 412 m²), we use a square loop of equal area (20.4 m x 20.4 m) in the inversion.

M: We added the above listed details on the TEM data and inversion to the methods sections and the new appendix.

Moreover, there is currently no data visualized (Only the TD-IP lab data). I suggest to include a section with data, and possibly also with inversion model response (If feasible, for example in an appendix). Of course this should not distract from the study itself.

A: We see the importance of including visualizations of exemplary data and inversion model responses (as an appendix) in order to allow the interested reader to quickly gain an impression of data quality and inversion model fits. Besides this new appendix, all data, inverted models and computed responses will still be available from the open data repository and can be revised there in detail.

M: We added an appendix to show exemplary data and inversion model responses for TEM, TDIP, and SRT measurements.

• From my experience, the TEMfast device sometimes shows significant distortions using small loop configurations. Did you observe any data distortions especially since a very small configurations was used? And, did you for example compare some land based soundings using a larger transmitter to validate that the very small layout gives correct transient data? In this respect, I also suggest to show at least some data.

A: The new Fig. A1 shows measured and calculated apparent resistivity curves of 5 soundings along TEM Profile 5 of Lake Tzibaná (smooth and layered resistivity models shown in Figure 10 of the manuscript). The measured curves are well recovered by the inverted models and do not show any conspicuous features, which would point to a distortion (e.g., due to the small loop configuration).

We have not carried out test measurements with different loop sizes during the field work in Mexico. But we do have some test measurements at different locations in Europe: Figure R2.4 shows the impulse responses for single-loop measurements with 4 different sizes of the square loop (6, 12, 25, and 50 m). For times >10 μs, all loop sizes result in consistent transients without conspicuous distortions. In particular, the same applied for the uniform time window between 20 and 200 μs, to which our transients from the Mexican lakes were truncated. It is worth mentioning that the average resistivity of the test location (approx. 120 Ωm across the first 50 m) is slightly larger than the average resistivity in the lake environment (20-30 Ωm). However, from our own practical experience, distortions due to small loop sizes rather decrease when the ground is more conductive.

[Figure]

**Figure R2.1.** *Impulse response of a single-loop configuration using a TEMfast device at a side with an average resistivity of 120 Ωm across the uppermost 50 m (Donau Island, Vienna, Austria). The solid lines show the impulse responses for square loops with side lengths of 6 m (dark blue), 12 m (light blue), 25 m (yellow), and 50 m (brown). The dotted lines show the corresponding error levels (determined as the standard deviations of the repeated measurements). The red rectangle highlights the time window between 20 – 200 μs, to which the transients of our study at the Mexican lakes were truncated uniformly.*

M: In the discussion section, we added a short comment on possible distortions due to small loop configurations and shortly discuss the (probable) absence of such adverse effects in our data set. The exemplary data and inversion model responses for TEM measurement in the new appendix furthermore enable the reader to follow this discussion and to visually assess the data quality.

We have not included the above test measurements with different loop sizes in order to avoid overloading the manuscript.

• A conductor is indicated below the limestone towards the east in Fig. 7a. Please discuss this feature if it can be related to any geology such as fracture zones or if this is an artifact (probably related to distorted late time transient data). A slightly similar feature is also seen in Fig. 9 towards the south.

A: Unfortunately, both conducting features (old Figures 7a and 9a) are located at the very ends of the TEM lines, which are not covered by the collocated TDIP profiles. In addition, geological reference data, such as detailed maps or even drillings, are not available for these depths. Thus, we do not have any control on the nature of these features, i.e., we cannot decide whether they reflect a geological feature (e.g., a fracture zone or a more conductive claystone unit) or arise from distorted late time data.

M: We included a discussion of this issue into the revised version of the manuscript.

• Does the ZOND software actually invert for coincident loop or for a central loop receiver? For very early times the central loop transients differ from coincident loop data.

A: In order to be sure, we have checked this detail with the author of the code (Alex Kaminsky, zondgeo@gmail.com). His response can be summarized as follows: The software ZondTEM1d supports any arbitrary shaped loops, the vertices of which can be defined independently. The response is calculated as the integral along the transmitter loop path, i.e., the transmitter loop is implemented as a set of directed electrical dipoles. The same applied to the receiver loop, where the response (Bz) is integrated over the exact area of the receiver area.

M: We briefly mention this detail in the revised methods section.

• The TEM data might be effected by 2D effects especially considering rather steep slope angles towards the edges. possibly include some discussion such as "multidimensional effects in TEM data were not considered as the TEM survey lines were not along strong bathymetry or steep slopes".

A: We are aware of the possible problems of multidimensionality can cause in the 1D interpretation of TEM soundings on lakes with steep bathymetry gradients (see, e.g., the extensive discussion of this issue provided by Mollidor et al., 2013). As you mention, the variation of the bathymetry along the lines shown here is relatively slight, which implies that this type of problem will probably not be of great relevance here. The assumption of onedimensionality might be more problematic at sounding sites located close to the lake shore.

M: We included a corresponding note in the methods section and come back to this topic in the discussion of the two profiles with TEM results to confirm that the bathymetry only varies softly along both lines and the assumption of one dimensionality is suitable, here.

• P315 - Obviously the p-wave velocity is less than expected. Can you elaborate why a lower vp < 2000 m/s was observed in the SRT measurements.

A: Possibly a misunderstanding? The depth of investigation of this SRT line (approx. 40 m below the water table of March 2018) is less than the depth of the surface of the limestone bedrock inferred from the TDIP resistivity section (> 40 m). Thus, the p-wave velocity at the lower limit of the SRT image is not lower than expected but lower than the typical p-wave velocity in the limestone unit.

M: We substitute the formulation "expected for limestone bedrock" by "typical for the limestone bedrock" to prevent this misunderstanding to happen.

• P350 - I suggest to include a table that summarizes the specifications of each method such as resolved physical parameter, DOI, pro/con of each method. Such a table would also summarize the used methods a bit and emphasize the integrative approach.

M: We liked this idea very much and included the new Table 2 into the revised manuscript, which summarizes the characteristic, the scope and limitations of all 4 field methods, i.e. SBP, TEM, TDIP, SRT.

• P-365 - For TEM a water-depth of 20 m depending on the water conductivity is not necessarily a limitation. Please correct this statement.

A: We fully agree: Due to the maximum water level reduction of about 20 m (from March 2018 to October 2019), we were only able to collect additional data on the dry lake floor at locations with less than 20 m of water column during our water-borne measurement campaign. This is the water depth down to which we were able to have a direct comparison of water-borne TEM data and

terrestrial TDIP data. Of course, this does not imply that the system does not also work in deeper water.

M: We added the following sentence to prevent possible misunderstandings: "Furthermore, there is no reason to assume that the system should not work as well in even deeper (>20 m) water depending on the water conductivity."

• For all interpretation a smoothness constraint inversion is used. Do you expect a smooth transition from the sedimentary layers to the limestone. In this respect, is a smoothness constraint inversion appropriate to image the geological situation here?

A: Actually, we have been thinking about including seismic contacts (from SBP images) as a-priori information (geometric constraints) into the TDIP and SRT inversion process. However, we decided not to further pursue this approach as we consider it more conclusive to have various methods confirming similar structures without "forcing" geometries to coincide by using constraints in the inversions.

While we do not see any real alternative to a smooth inversion of our 2D data sets (i.e., TDIP and SRT), we do agree that it is not as straight forward to only discuss the results of a smoothness constrained approach for the TEM inversion. Here, the decision to only show the smooth models was motivated by facilitating the intended comparison with the (smooth) TDIP resistivity images.

In addition, there is no simple answer to the question whether we expect a sharp resistivity contrasts between the main lithological units, i.e.., sediment cover and limestone, or not. As our interpretation of the Metzabok data suggests, within the mixed layer (sediment and limestone debris/heavily fractured limestone) there might well be a rather smooth transition as a result of a continuously increasing volume content of limestone with depth. The same is true for the contacts between the different sedimentary units (fine-grained lake sediments/sandy delta deposits), which we expect to be rather gradual, too.

M: We included layered models into the visualization of the TEM resistivity models (i.e., Fig. 7a and 10a of the revised manuscript). We also include a short discussion of these two inversion approaches into the revised discussion section.

• As water-borne TEM studies are still quite rare, I miss some references to recent water borne TEM studies. For example, we recently applied boat-towed TEM to image a hydrothermal target on the Azores. In this study we gathered around 600 soundings using the TEM system (initially developed by Mollidor et al.) in a continuous mode. There are also other very recent studies. These can be included as references, if the authors find them suitable:

- Yogeshwar, P., Küpper, M., Tezkan, B., Rath, V., Kiyan, D., Byrdina, S., ... & Viveiros, F. (2020). Innovative boat-towed transient electromagnetics Investigation of the Furnas volcanic lake hydrothermal system, Azores. Geophysics, 85(2), E41-E56.

- Lane Jr, J. W., Briggs, M. A., Maurya, P. K., White, E. A., Pedersen, J. B., Auken, E., ... & Adams, R. (2020). Characterizing the diverse hydrogeology underlying rivers and estuaries using new floating transient electromagnetic methodology. Science of The Total Environment, 140074.

A: Thanks a lot for the hint! We missed these very recent studies. They are more than relevant.

M: We updated the state-of-the-art part on boat-towed TEM devices accordingly and included these additional references.

**3. Technical corrections**

The MS is very well written and the language is very good. All figures are well prepared with well readable fonts. Therefore, I only have a few technical corrections:

• P70 - the term reference data is misleading. I do not see that the data is actually used as reference data. Better - "additional/complementary data for comparison with the water borne data"

M: We replaced the term "reference data" by "additional data".

• Please check that all abbreviations are defined, e.g. ERT etc.

A: We have checked the abbreviations. Besides the undefined abbreviation ERT (line 77 and in the caption of Figure 6), we have not found any additional problems with undefined abbreviations.

M: We replaced the term "ERT results" in line 77 by "TDIP resistivity results" and the term "ERT/IP data" in the caption of Figure 6 by "TDIP resistivity and phase data".

• P140 - explain or remove the skip parameters (skip-1 skip-2 etc.)

M: We removed the skip parameters specified between the brackets.

[revised manuscript text omitted]

limestone hillocks    seismic reflector    erosional depression    shore

limit of conductive units    karst conduits

Water (lake and karst aquifer)    Collapse debris and fine-grained sediment
Fine-grained lake sediments    Limestone with sediment-filled fractures
Limestone bedrock    Limestone with water-filled fractures

(a) limestone hillocks    seismic reflector    erosional depression    shore

limit of conductive units    karst conduits

Water (lake and karst aquifer)    Collapse debris and fine-grained sediment
Fine-grained lake sediments    Limestone with sediment-filled fractures
Limestone bedrock    Limestone with water-filled fractures

(b)  (c)  (d) Fractured limestone

~1.5 m

Fine-grained sediment

[revised manuscript text omitted]

---

## Referee Report (RR1)

**1. General Comments**

I have carefully re-reviewed the manuscript (MS).

For my feeling, the authors have significantly improved the MS. The red line has become very clear and is worked out well now in the revised version. Furthermore, the authors have expanded the results and included various discussion points also to give a more general conclusion. The reader is guided clearly through the study now.

Actually all my points were discussed in great detail and all suggestions were considered and followed where it was appropriate. I appreciate the very precise and clear answers of the authors in this revision round. Therefore, I do not have any more general comments.

**2. Specific comments**

All technical comments where considered and answered in detail. Where appropriate the text was expanded and discussion points included. However, I do have a few more specific comments/suggestions that can be considered:

- I do suggest to included a small overview Figure 1a with the boarders of Mexico. It is more convenient to start location description at a larger scale.

- I found it quite interesting that the lakes have differing water conductivity. The chemical analysis of the sediments and the water are discussed in the MS later on. Maybe you can add some sentence if they do shed some light on the evolution of the lakes.

**3. Technical corrections**

Some few technical corrections remain:

- L260 - TU - Vienna
- L300 - local heights -> maybe better "...local rising of the ..."
- L315 - possibly include one sentence, e.g. "... which shows the benefit of evaluating the TD-IP phase".
- Figure 4 - it would be actually nice if the width of the subfigures are adjusted to that the profile meters are aligned to compare the structures better in a 1-1 way.
- Figure 8 b-d - include some reference in the text and short description. Please check also for other subfigures if they are referenced in the text.
- p14 - p16 - please check language. For me the language here can be improved, e.g. "underpinned", "were unplugged".

With best wishes, Pritam Yogeshwar

---

## Author Response (AR2)

In the following, reviewer comments are typeset in black and the authors' responses in blue.

**Author response to Referee #1 (anonymous)**

I am very impressed with the new version of the manuscript. The overall objectives of the paper are now much clearer and demonstrate the value of the present work much more clearly. The paper shows in an amazing way how the integrative use of different geophysical methods can provide a clearer picture of the conditions in the subsurface and thus also shed light on the processes taking place. In addition to the local results, the study also provides an excellent summary of the possibilities and limitations of the methods used.

A: We thank the reviewer for having a second close look into our revised manuscript. We are particularly happy to know that our modifications meet the high expectations of the reviewer. We would like to reiterate our gratitude for the detailed feedback provided during the first review, which was a great help to substantially improve the manuscript.

**Author response to Referee #2 (Pritam Yogeshwar, University of Cologne)**

**1. General Comments**

I have carefully re-reviewed the manuscript (MS).

For my feeling, the authors have significantly improved the MS. The red line has become very clear and is worked out well now in the revised version. Furthermore, the authors have expanded the results and included various discussion points also to give a more general conclusion. The reader is guided clearly through the study now.

Actually all my points were discussed in great detail and all suggestions were considered and followed where it was appropriate. I appreciate the very precise and clear answers of the authors in this revision round. Therefore, I do not have any more general comments.

A: We thank Pritam Yogeshwar a lot for his continued work on this manuscript and the additional comments and corrections, which we address point by point below. Together with the detailed and constructive feedback provided during the first round of reviews, these hints helped us to further improve the manuscript! Thank you!

**2. Specific comments**

All technical comments where considered and answered in detail. Where appropriate the text was expanded and discussion points included. However, I do have a few more specific comments/suggestions that can be considered:

• I do suggest to included a small overview Figure 1a with the boarders of Mexico. It is more convenient to start location description at a larger scale.

**A: We included an overview map as Fig. 1a showing the location of the study area with respect to the political boarders of Mexico and the neighboring Central American countries.**

• I found it quite interesting that the lakes have differing water conductivity. The chemical analysis of the sediments and the water are discussed in the MS later on. Maybe you can add some sentence if they do shed some light on the evolution of the lakes.

A: The variation of water conductivity in the lakes still remains an open question. Limnological research to improve the understanding of these variations is underway.

In order to account for this comment, we added the following sentence (L245 of the revised, marked manuscript): "However, a comprehensive understanding of the strong variation of water conductivity with respect to both sampling location and time is subject of ongoing limnological research in the study area."

**3. Technical corrections**

Some few technical corrections remain:

**• L260 - TU – Vienna**

A: The institution's own guidelines require to spell the name as it is "TU Wien". However, we inserted "Vienna" as name of the city after the institution's name.

• L300 - local heights -> maybe better "...local rising of the ..."

**A: Done.**

• L315 - possibly include one sentence, e.g. "... which shows the benefit of evaluating the TD-IP phase".

A: Done. We included the sentence "The capability to separate the pure sediments from the limestone-sediment mixture underlines the benefit of evaluating the TDIP phase."

• Figure 4 - it would be actually nice if the width of the subfigures are adjusted to that the profile meters are aligned to compare the structures better in a 1-1 way.

**A: Done as suggested.**

• Figure 8 b-d - include some reference in the text and short description. Please check also for other subfigures if they are referenced in the text.

A: We included references to Figure 8b-d in the main text and added a sentence referring to the mixed material shown in Fig. 8 b and c (L345 and L349 of the revised manuscript).

We also checked whether all subfigures were called out – either separately or as a part of the entire figure – in the main text. Where ever necessary, we included new or corrected erroneous references.

• p14 - p16 - please check language. For me the language here can be improved, e.g. "underpinned", "were unplugged".

A: In this context, we would rather substitute the word "underpinned" by "supported", which we did in the revised version of the manuscript.

We also checked the remaining parts of the new paragraphs on the indicated pages and made the following additional changes to improve the language (line numbers of the revised, marked manuscript):

L403: "more considerable" -> "considerably higher"

L410: "further of the shore" -> "farther offshore"

L414: "Thus, where **it** is thick enough (up to 5-6 m across large areas), **this layer** acts..." -> "Thus, where **this layer** is thick enough (up to 5-6 m across large areas), **it** acts..."

L444: water**b**borne -> waterborne

L463: "such" deleted

L475: "alone" added

L483: "image" -> "locate"

L489: "5.3.2" -> "5.3.3"

L540: Reorganized sentence: "The possibility to recollect additional data directly on the exposed lake floor after the sudden drainage of Lakes Metzabok and Tzibaná substantially benefitted the evaluation of the different methods."

L550: "only" -> "alone"

**Other relevant modifications**

Added reference to Przyklenk et al. (2016) in L135 and included bibliographic information in the references section (L709).

All modifications made during this revision are marked in red on the following pages.

**Integrated land and water-borne geophysical surveys shed light on the sudden drying of large karst lakes in southern Mexico**

Matthias Bücker1,2, Adrián Flores Orozco2, Jakob Gallistl2, Matthias Steiner2, Lukas Aigner2, Johannes Hoppenbrock1, Ruth Glebe1, Wendy Morales Barrera3, Carlos Pita de la Paz4, César Emilio García García4, José Alberto Razo Pérez4, Johannes Buckel1, Andreas Hördt1, Antje Schwalb5, Liseth Péerez3,5

[revised manuscript text omitted]
| TEM    | $ ho^{**}$ 5–500 $\Omega$ m                               | Water borne;
unanchored single-loop
system, 412 m 2 loop
area, 1 A transmitter
current, rubber boat | 50–100     | Delineates top of
bedrock (e.g., Fig. 7a)
and coarse delta deposits
(e.g., Fig. 10a)                                                                         |  <li>Low acquisition
velocity/productivity
compared to SBP</li>                                                                                             |
| TDIP   | $\rho$
5–500 $\Omega$ m
$\varphi^{***}$
2–6 mrad | Terrestrial; 5–10 m
spacing, 48 electrodes,
dipole-dipole, 0.5–1 A
transmitter current,
500 ms pulse           | 50–70      |  <li>ρ: Delineates coarse delta deposits</li> <li>(e.g., Fig. 10b)</li> <li>φ: Improved delineation of fine-grained sediments</li> <li>(e.g., Fig. 5b)</li>  | – No clear distinction
between lake sediments
and limestone bedrock if
only $\rho$ is considered
(e.g., Fig. 4f)                                                 |
| SRT    | ν p
200 – 3000 m/s                          | Terrestrial; 5 m
spacing, 24 geophones
(28 Hz), energy source:
7.5 kg sledge hammer                               | 40–50      | Delineates top of
bedrock (e.g., Fig. 7d)                                                                                                                          |  <li>Eventually low quality</li> <li>of data acquired on muddy</li> <li>lake floor</li>                                                                             |

760 \*p-wave velocity, \*\*electrical resistivity, \*\*\*resistivity phase